# *Borrelia burgdorferi* lacking all cp32 prophage plasmids retains full infectivity in mice

Chad Hillman[1], Hannah Theriault[1,3,4], Anton Dmitriev[2,5], Satyender Hansra[2], Patricia A Rosa[1] & Jenny Wachter [ID][1,6 ✉]

## Abstract

**The causative agent of Lyme disease, *Borrelia burgdorferi*, contains a unique, segmented genome comprising multiple linear and circular plasmids. To date, the genomes of over 63 sequenced Lyme disease *Borrelia* carry one or more 32 kbp circular plasmids (cp32) or cp32-like elements. The cp32 plasmids are endogenous prophages and encode, among other elements, a family of surface exposed lipoproteins termed OspEF-related proteins. These lipoproteins are synthesized during mammalian infection and are considered important components of the spirochete's adaptive response to the vertebrate host. Here, we detail the construction and infectivity of the first described *B. burgdorferi* strain lacking all cp32 plasmids. Despite their universal presence, our findings indicate that *B. burgdorferi* does not require any cp32 plasmids to complete the experimental mouse-tick-mouse infectious cycle and a total lack of cp32s does not impair spirochete infectivity.**

**Keywords** Lyme Disease; *Borrelia burgdorferi*; cp32 Plasmids; Complement Resistance
**Subject Categories** Genetics, Gene Therapy & Genetic Disease; Immunology; Microbiology, Virology & Host Pathogen Interaction

## Introduction

*Borrelia burgdorferi*, the causative agent of Lyme disease, is transmitted to susceptible vertebrate hosts through the bite of an infected *Ixodes* tick (Burgdorfer et al, 1982; Lane et al, 1993; Steere et al, 1983). Prior to transit from the *Ixodes* tick vector, *B. burgdorferi* must sense the environmental changes that accompany the incoming blood meal and initiate a global response through the alternative sigma factor RpoS as a prerequisite for host infection (Balashov, 1972; Carroll et al, 1999; de Silva et al, 1996; Elias et al, 2002; Fingerle et al, 1998; Fisher et al, 2005; Hodzic et al, 2002; Kasumba et al, 2016; Kurokawa et al, 2020; Ohnishi et al, 2001;

Ribeiro, 1988; Schwan et al, 1995). While *rpoS* is encoded on the chromosome, many genes important for spirochete survival throughout the enzootic cycle are plasmid-borne.

*B. burgdorferi* contains a unique, highly segmented genome comprising a linear chromosome and more than 20 distinct, co-existing circular and linear plasmids (Barbour and Garon, 1987; Baril et al, 1989; Casjens et al, 2000; Casjens et al, 2017; Ferdows and Barbour, 1989; Mongodin et al, 2013). Among these genetic elements are multiple prophage plasmids whose genes are positively regulated by RpoS (Caimano et al, 2019; Caimano et al, 2007; Wachter et al, 2023). These prophage plasmids in the type strain B31 include multiple co-existing forms of the circular plasmid (cp) 32 prophage, as well as a cp32 prophage-like element on linear plasmid (lp)56, interspersed cp32-like phage genes on lp54, and an unrelated linear prophage element on lp28-2 (Casjens et al, 1997; Eggers et al, 2006; Jutras et al, 2013; Schwartz et al, 2021; Yang et al, 2003; Zhang and Marconi, 2005). To date, >63 sequenced and annotated Lyme disease spirochetes (*B. burgdorferi* sensu stricto and sensu lato strains) contain at least one, and as many as 10, cp32 plasmids or cp32-like elements in their genomes (Casjens et al, 2011; Mongodin et al, 2013). The only exception is a single isolate of *B. garinii* Far04 that does not contain canonical cp32 plasmids, but rather contains a linear lp32-10 plasmid that encodes the *parA* plasmid replication gene of cp32-10 but differs from the gene content of typical cp32 plasmids (Casjens et al, 2011). While multiple phages have been detected in association with *B. burgdorferi*, their contributions throughout the infectious cycle have not been determined (Eggers et al, 2016; Eggers et al, 2001; Eggers and Samuels, 1999; Hayes et al, 1983).

As lysogenic prophage, the cp32 plasmids encode a family of surface-exposed proteins referred to as OspEF-Related Proteins (Erps) (Akins et al, 1999; Casjens et al, 1997; Marconi et al, 1996; Stevenson et al, 1998; Stevenson et al, 1996). These *erp* genes contain similar promoter regions and are expressed in culture under conditions that mimic the mammalian environment (Akins et al, 1995; Babb et al, 2001; Hefty et al, 2001; Ojaimi et al, 2003; Stevenson et al, 1995), as well as during mammalian infection (McDowell et al, 2001; Miller and Stevenson, 2006; Miller et al, 2006). In fact, most cp32-encoded genes, not just the *erps*, are positively regulated by RpoS, consistent with their expression in the

[1]Laboratory of Bacteriology, National Institute of Allergy and Infectious Diseases, National Institutes of Health, Hamilton, MT, USA. [2]Vaccine and Infectious Disease Organization, University of Saskatchewan, Saskatoon, SK, Canada. [3]Present address: Department of Biomedical Sciences, School of Public Health, State University of New York, Albany, NY 12144, USA. [4]Present address: The Arbovirus Laboratory, New York State Department of Health, Wadsworth Center, Slingerlands, NY 12159, USA. [5]Present address: Temerty Faculty of Medicine, University of Toronto, Toronto, ON, Canada. [6]Present address: Vaccine and Infectious Disease Organization, University of Saskatchewan, Saskatoon, SK, Canada. ✉E-mail: jenny.wachter@usask.ca

vertebrate host (Caimano et al, 2019; Caimano et al, 2007; Eggers et al, 2006; Jutras et al, 2013; Yang et al, 2003). Erp proteins are purported to act as adhesins and facilitate spirochete survival and attachment in vertebrate hosts. Erps can be divided into three subfamilies based on mature protein sequence homology and predicted binding moieties. These include the OspE-related, OspF-related, and OspEF-Like Leader Peptide (Elp) proteins (Akins et al, 1995; Caimano et al, 2000). The Elp proteins share similar leader peptides as OspE/OspF but have completely unrelated mature protein sequences. OspE-related Erp proteins, ErpA and ErpP, along with lp54-encoded CspA and lp28-3-encoded CspZ, are believed to prevent complement activation by binding the complement regulator Factor H (Hartmann et al, 2006; Kraiczy et al, 2003; Lin et al, 2020; Stevenson et al, 2002). These *Borrelia burgdorferi* Complement-Regulator Acquiring Surface Proteins (BbCRASPs) have been shown to differ in expression patterns throughout the enzootic cycle, suggesting distinct roles in immune evasion (Bykowski et al, 2007; Hart et al, 2018b; Lin et al, 2020; Marcinkiewicz et al, 2019). Additionally, ErpA and ErpP are hypothesized to facilitate spirochete dissemination through plasminogen binding (Brissette et al, 2009b). While the majority of OspE/OspF protein targets have been determined through in vitro binding assays, there have been some in vivo studies. In a *B. burgdorferi* derivative lacking lp28-4 and lp56, it was found that an *erpA* transposon mutant exhibited delayed dissemination in murine tissues (Lin et al, 2012). In *B. burgdorferi* lacking plasmid lp21, CspA-deficient spirochetes were fully infectious in mice by needle-inoculation, but were cleared in feeding nymphs and failed to be transmitted to naive hosts (Hart et al, 2018a). Additionally, while CspZ was found to protect an attenuated *B. burgdorferi* derivative from serum in vitro (Hartmann et al, 2006), *cspZ*-mutants in a wt background were fully infectious in mice with no defects in tissue loads (Coleman et al, 2008). Some OspF-related proteins, including ErpG, ErpL, ErpY, and ErpK, have been shown to bind heparan sulfate, with reduced colonization of the joints, heart, and skin in spirochetes that lack plasmids lp28-4 and lp56 and contain a transposon mutant in *erpK* (Antonara et al, 2007; Lin et al, 2012; Lin et al, 2015). Finally, ElpB and ElpQ (also known as ErpB and ErpQ), along with lp36-encoded BBK32, bind and inhibit the activity of the complement C1 complex in vitro (Garcia et al, 2016; Garrigues et al, 2022; Pereira et al, 2022). While these studies highlight the potential roles of Erp proteins in vivo, it is unclear if the phenotypes described reflect the loss of plasmids (lp28-4 and lp21) in addition to the Erps, or can be ascribed solely to the Erps. It has been reported that outer surface proteins of *B. burgdorferi* have multifunctional and redundant roles to promote survival and tissue tropism (Caine and Coburn, 2016). However, it has been difficult to elucidate the functions of individual Erp/Elp proteins due to the multiple copies of *ospE/F* genes in *B. burgdorferi*.

Recently, we demonstrated that inhibiting expression of the RpoS regulator, *bbd18*, results in an unmodulated increase in *rpoS* expression, which in turn leads to induction of cp32 prophage and culminates in cell lysis in vitro (Wachter et al, 2023). To determine if phage were responsible for the observed cell death, we systematically displaced all seven cp32 plasmids from a *B. burgdorferi* strain B31 derivative. This was the first description of a *B. burgdorferi* strain lacking lp56 and all known cp32 plasmids, hence we sought to further characterize this strain and assess the phenotype of these spirochetes throughout the experimental mouse-tick infectious cycle (Mongodin et al, 2013). Thus far, the ability to study

certain cp32-encoded genes has been hindered by the redundancy of multiple copies of cp32 plasmids and genes co-existing in a single *B. burgdorferi* strain. Herein, we report that *B. burgdorferi* lacking all cp32 plasmids can successfully complete the infectious cycle, with no defect in transmission and infection of the murine host or uptake and retention in the tick vector. These derivatives will allow for further investigation of cp32-encoded genes to provide insights into their functions throughout the infectious cycle. Additionally, in the absence of the cp32 plasmids, we did not find an increase in expression of non-cp32 genes encoding proteins of proposed similar functions at any stage of the mouse-tick infectious cycle. We conclude that the Lyme disease spirochete does not require the cp32 plasmids, or any encoded Erp/Elp proteins, to establish infection, persist, and be transmitted between the tick vector and murine host.

# Results and discussion

## Displacement of seven cp32 plasmids

We previously demonstrated that over-expression of *rpoS*/RpoS led to spirochete lysis, with concomitant increase in cp32 plasmid copy number and cp32 transcripts, and the presence of phage-like particles in the supernatant of lysing cultures (Wachter et al, 2023). In order to determine if cp32 phage induction was the cause of cell death following *rpoS*/RpoS over-expression, all seven cp32 plasmids were systematically displaced from *B. burgdorferi* strain B31-A3-68-LS (wt) (Fig. 1A,B) (Wachter et al, 2023). This was accomplished by designing a series of shuttle vectors (SV) that contain an incomplete set of plasmid maintenance genes (pfam-57, 30, and 32) for each cp32 plasmid of strain B31. These SVs, which are replication-competent in both *E. coli* and *B. burgdorferi*, and incompatible with the native cp32 plasmid from which they are derived, were sequentially transformed into *B. burgdorferi* to displace the endogenous cognate cp32. While these SVs can replicate in *B. burgdorferi*, they lack a *parB*/Pfam 49 gene, which renders them unstable and readily lost in the absence of antibiotic selection. This feature permitted sequential displacement of all seven cp32 plasmids in strain B31 using the same *E. coli* plasmid backbone and a single selectable marker (Fig. 1A,B).

Through sequential plasmid displacement, we have created the first strain B31 derivative lacking lp56 and all seven cp32 plasmids, referred to hereafter as Δcp32. We previously determined that loss of all seven cp32 plasmids from wt *B. burgdorferi* strain B31-A3-68-LS (Δcp32) does not impair growth in culture (Wachter et al, 2023), with the analysis of each resulting derivative also revealing no growth defect in vitro following loss of one or more cp32 plasmids (Fig. 1C). However, because the expression of cp32-encoded genes is regulated by RpoS and as some of the cp32-encoded proteins are believed to be important during mammalian infection, we undertook to assess the virulence of Δcp32 spirochetes in the murine host and tick vector. Therefore, in this study we characterize the in vivo phenotype of the Lyme disease spirochete lacking all cp32 plasmids throughout the experimental mouse-tick-mouse infectious cycle.

## cp32 plasmids are dispensable throughout the infectious cycle

While the cp32 plasmids are well conserved among Lyme disease *Borrelia*, cp32 gene expression is quite low during in vitro

**A.** Creation of Δcp32 *Borrelia*

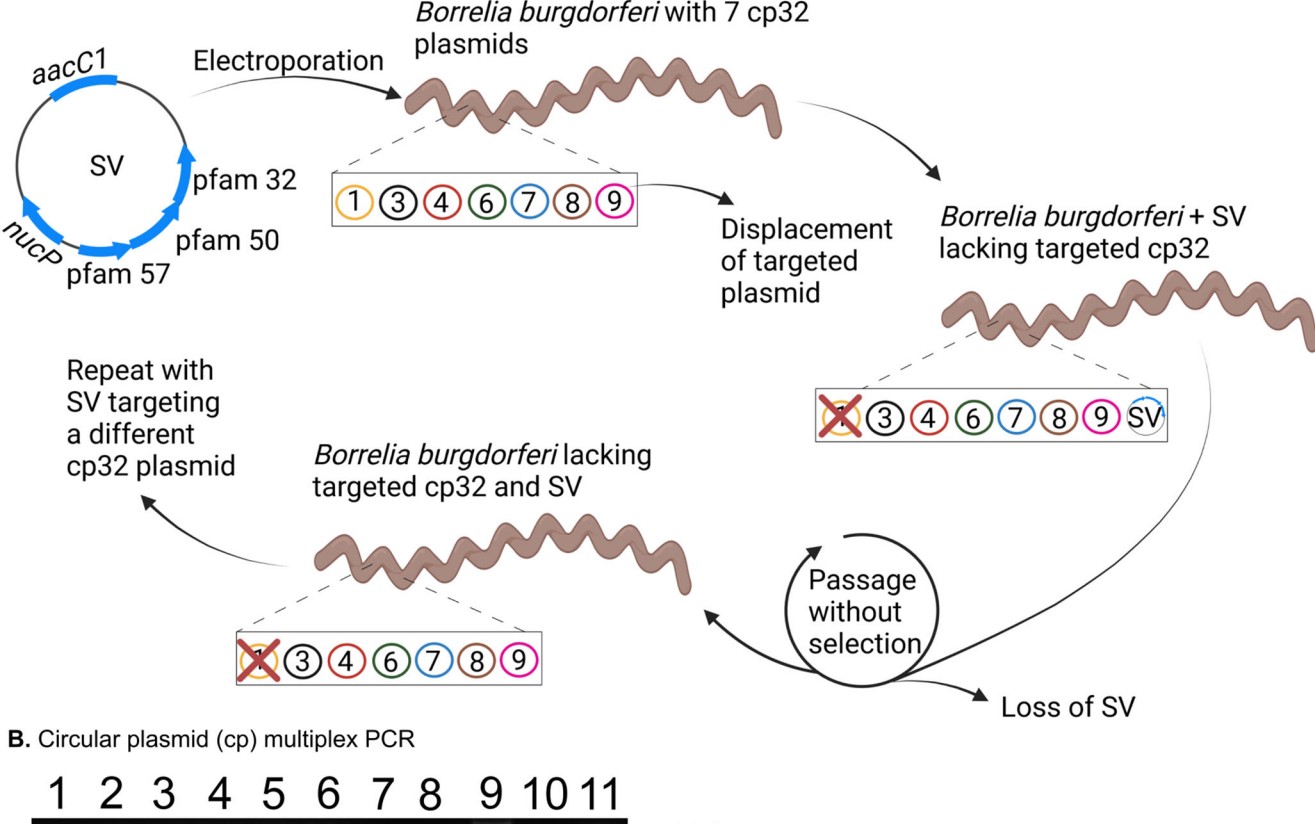

**B.** Circular plasmid (cp) multiplex PCR

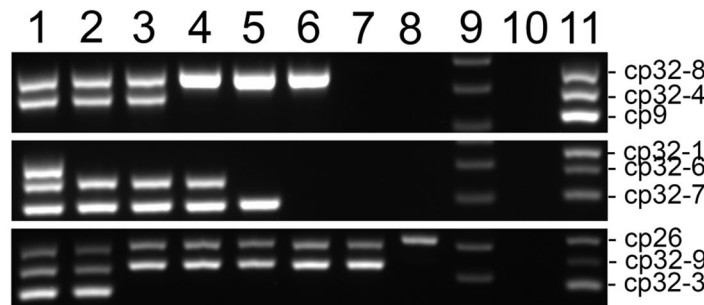

**C.** Growth curve of strains

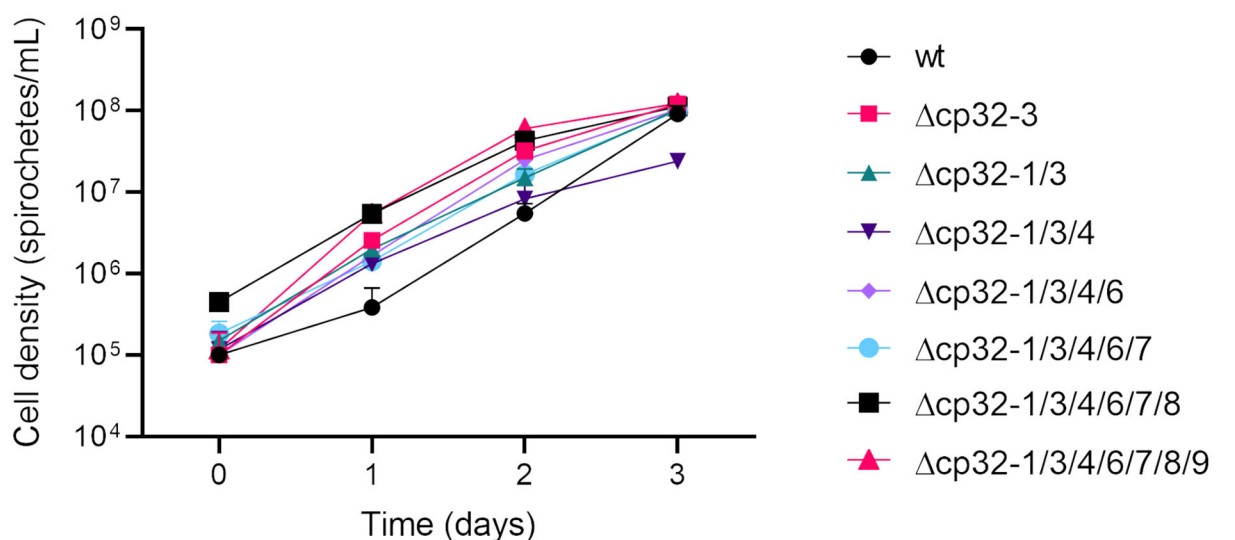

**Figure 1. Displacement of cp32 plasmids.**

(A) Schematic for generation of Δcp32 *Borrelia*. (B) Multiplex PCR for cp32 plasmids demonstrating sequential displacement of cp32 plasmids. Gel is loaded as: 1. wt (A3-68-LS); 2. Δcp32-1; 3. Δcp32-1/3; 4. Δcp32-1/3/4; 5. Δcp32-1/3/4/6; 6. Δcp32-1/3/4/6/7; 7. Δcp32-1/3/4/6/7/8; 8. Δcp32-1/3/4/6/7/8/9; 9. O'gene 1 kb + ruler; 10. No template PCR control; 11. *B. burgdorferi* B31-MI positive control. The plasmids used for shuttle vector design (A) and multiplex PCR primers (B) are described in Reagents and Tools Table. (C) Growth curve of strains reveals no negative phenotype in vitro due to loss of one or more cp32 plasmids. Data are presented as the mean values of three biological replicates +/− SD. n = 3 biological replicates. Source data are available online for this figure.

cultivation, which may explain why spirochetes lacking all cp32s did not display reduced growth in culture. However, as several cp32-encoded genes are purported to facilitate infection of vertebrate hosts, we sought to determine the viability of Δcp32 spirochetes throughout the mouse-tick infectious cycle. To this end, mice were needle-inoculated with either wt or Δcp32 *B. burgdorferi*. Contrary to expectations, Δcp32 spirochetes were able to persistently infect mice as confirmed by seroconversion at three weeks, and recovery of spirochetes from mouse tissues at 2-, 4-, and 5-weeks, post-injection (23/24 mice inoculated with wt and 28/29 mice inoculated with Δcp32). A subset of these outgrowths underwent multiplex PCR to confirm the circular plasmid profile of the *B. burgdorferi* isolates from murine ears (Fig. EV1) (Bunikis et al, 2011). This further confirmed that Δcp32 spirochetes are fully infectious and can be isolated from murine tissues at 2-, 4-, and 5-weeks post-inoculation. Additionally, the spirochete loads in tissues (ears, hearts, and joints) of needle-inoculated mice were analyzed by qPCR at 2-, 4-, and 5- weeks post-inoculation, with no significant differences in tissue load determined between wt and Δcp32 spirochetes at any time point (Figs. 2A,B and EV2).

To assess the phenotype of Δcp32 spirochetes in the tick vector, *Ixodes scapularis* larvae were fed to repletion on wt- and Δcp32-infected mice. When measured at 1, 7, and ~16 weeks post-repletion, spirochetes were recovered from most fed larvae (6/6 infected with wt and 7/8 infected with Δcp32 at 1 week) (Fig. 2D). The spirochete burdens in replete larvae were also similar between *I. scapularis* larvae infected with wt and Δcp32 *B. burgdorferi* (Fig. 2D). While the number of spirochetes was significantly higher in Δcp32-infected unfed nymphs compared to wt-infected, the spirochete burdens in replete nymphs at all other time points tested were similar (Fig. 2D). This outcome demonstrates that cp32 plasmids are not required by *B. burgdorferi* for acquisition by *Ixodes scapularis* larvae or transstadial transmission through the molt to the nymphal stage.

The retention of spirochetes by unfed nymphs allowed us to assess the ability of Δcp32 spirochetes to infect mice by tick bite, the natural mode of transmission. Therefore, wt- and Δcp32-infected unfed nymphs were fed to repletion on naive mice. Similar to infection by needle inoculation, wt and Δcp32 spirochetes were infectious for mice by tick bite, with the majority of mice seropositive at 3 weeks post-feeding and all mice carrying disseminated infections at 5 weeks post-feeding (3/3 mice fed upon by wt-infected nymphs and 6/6 mice fed upon by Δcp32-infected nymphs). Spirochete loads in tissues (ear and heart) were also analyzed by qPCR for mice infected with wt or Δcp32 spirochetes by tick bite at 5-weeks post feeding (Fig. 2C). Additionally, no difference in spirochete survival in fed ticks was detected when spirochete loads were assessed in fed nymphs at drop-off, 24 h, 48 h, 10 days and 1 month after feeding to repletion (Fig. 2D).

This outcome was unexpected, that spirochetes lacking all cp32 plasmids retain infectivity in both the murine host and tick vector with no statistically significant differences detected in tissue loads (Fig. 2). These data shows that the highly conserved and ubiquitous cp32 plasmids of Lyme disease *Borrelia* are not required for completing the experimental mouse-tick infectious cycle.

The wild-type phenotype of Δcp32 spirochetes during the experimental mouse-tick infectious cycle contradicts the proposed role of the cp32 plasmid-encoded *erp* and *elp* genes, which are thought to contribute to serum resistance and tissue tropism (Akins et al, 1999; Bykowski et al, 2007; Caimano et al, 2000; Hart et al, 2018b; Lin et al, 2020; Marcinkiewicz et al, 2019; Pereira et al, 2022). However, it's important to note that Δcp32 spirochetes still carry genes that encode proteins with similar functions, and their expression might compensate for the absence of the cp32 plasmids, thereby mitigating any potential negative effects.

## Gene expression of select genes in vitro and in vivo

The products of some cp32-encoded genes are proposed to facilitate infection of the mammalian host, yet Δcp32 spirochetes are competent to complete the experimental infectious cycle, possibly through functional redundancy of other gene products. We considered functional redundancy at the transcriptional level; if cp32-encoded genes were vital for immune evasion in immuno-competent hosts, then genes encoding complementary proteins of similar function should have increased expression to compensate for the loss of the cp32s. Therefore, to determine whether proteins with functions similar to the cp32-encoded gene products could compensate for their absence, we analyzed the expression of implicated genes in infected mouse hearts and in fed nymphs. The ascribed functions and proposed in vivo roles of genes encoded by the cp32s and non-cp32 genes analyzed herein are shown in Table 1.

There were no significant differences in transcript levels for all genes analyzed between Δcp32 and wt spirochetes in either infected heart tissues or replete nymphs (Figs. 3 and EV2). Unfortunately, we could not reliably detect *bba68, bb0347*, or *lmp1* transcripts in mice or nymphs infected with either wt or Δcp32 spirochetes (Fig. 3), so their relative abundance could not be compared.

Although we did not assess protein levels, we found no increase in the transcription of these functionally redundant genes by Δcp32 spirochetes in infected mouse tissues (Fig. 3). However, we cannot eliminate the possibility that post-transcriptional regulation of these genes or the activation of parallel pathways in Δcp32 spirochetes may facilitate innate immune evasion. Nonetheless, our findings conclusively demonstrate that cp32-encoded gene products are not required at any point of the experimental laboratory mouse-tick infectious cycle.

## A. Spirochete burden in needle-inoculated mice - 2 w

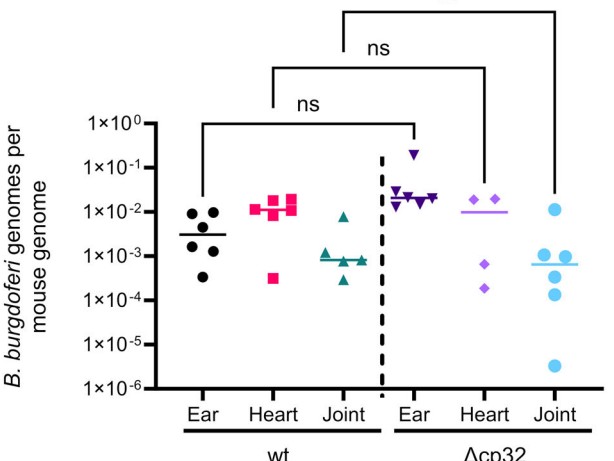

| Strain | Tissue | Number of tissues analyzed | Time point of tissue collection |
|--------|--------|----------------------------|--------------------------------|
| wt | Ear | 7 | 2 weeks |
| wt | Heart | 5 | 2 weeks |
| wt | Joint | 4 | 2 weeks |
| Δcp32 | Ear | 6 | 2 weeks |
| Δcp32 | Heart | 4 | 2 weeks |
| Δcp32 | Joint | 6 | 2 weeks |

## B. Spirochete burden in needle-inoculated mice - 5 w

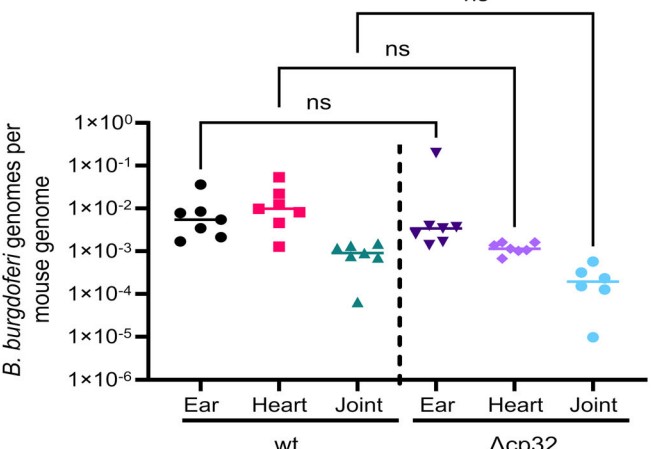

| Strain | Tissue | Number of tissues analyzed | Time point of tissue collection |
|--------|--------|----------------------------|--------------------------------|
| wt | Ear | 7 | 5 weeks |
| wt | Heart | 7 | 5 weeks |
| wt | Joint | 7 | 5 weeks |
| Δcp32 | Ear | 7 | 5 weeks |
| Δcp32 | Heart | 7 | 5 weeks |
| Δcp32 | Joint | 6 | 5 weeks |

## C. Spirochete burden in mice infected by tick bite

| Strain | Tissue | Number of tissues analyzed | Time point of tissue collection |
|--------|--------|----------------------------|--------------------------------|
| wt | Ear | 2 | 5 weeks |
| wt | Heart | 4 | 5 weeks |
| Δcp32 | Ear | 2 | 5 weeks |
| Δcp32 | Heart | 5 | 5 weeks |

## D. Spirochete burden in ticks

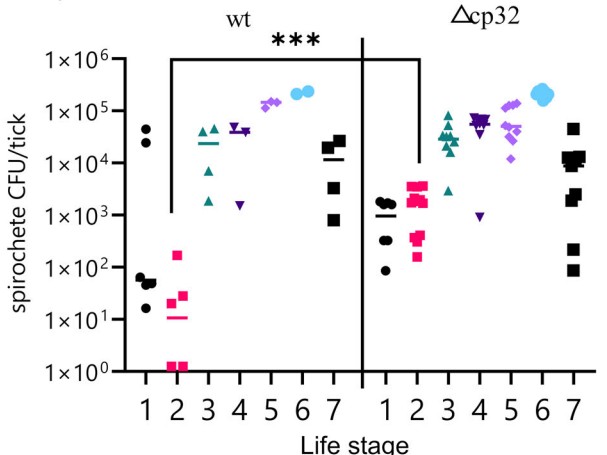

| | Number of infected ticks/ number of ticks assessed | |
|--------|------|-------|
| Life stage | wt | Δcp32 |
| 1. Replete larvae | 6/6 | 7/8 |
| 2. Unfed nymph | 5/6 | 12/12 |
| 3. Replete nymph | 4/4 | 9/10 |
| 4. Nymph 24 hrs | 3/3 | 9/10 |
| 5. Nymph 48 hrs | 3/3 | 10/10 |
| 6. Nymph 10 days | 2/3 | 9/10 |
| 7. Unfed adult (1 mo) | 4/4 | 9/9 |

◄ **Figure 2. Experimental infection and transmission of Δcp32 spirochetes.**

(A–C) Infectivity of wt and Δcp32 spirochetes in mice. Similar numbers of wt and Δcp32 genomes were detected by qPCR in ear, heart, and joint tissues of needle inoculated mice at 2-weeks (A) and 5-weeks post infection (B) and in ear and heart tissues of mice infected by tick bite (C). The number of tissues that were analyzed and the time points of tissue collection are shown as tables beneath their respective graphs. (D) Larval ticks acquired wt and Δcp32 spirochetes from needle-inoculated mice and maintained them through nymphal and adult stages. Viable wt and Δcp32 spirochetes were quantified as CFU in replete larvae, unfed nymphs, and replete nymphs at drop-off, 24 h, 48 h, 10 days, or 1 month following the nymphal blood meal. Comparable numbers of viable wt and Δcp32 spirochetes were detected in ticks at each time point except in unfed nymphs, which likely reflects the slight difference in replete larval loads. Data information: Data are presented in graphs as individual values of biological replicates, with the mean value indicated. Dunn's multiple comparison of the Kruskal–Wallis test was used to determine significance. (B) $n = 7$ biological replicates with 4–7 yielding useable data. (C) $n = 7$ wt biological replicates and 8 Δcp32 biological replicates with 1–8 yielding useable data. ***$p$-value < 0.0005 Source data are available online for this figure.

The in vivo data from this study, as well as published data cited in Table 1, demonstrate the shortcomings of overinterpreting in vitro binding assays to infer the vital in vivo functions of genes. In fact, despite the theoretically important roles of proteins encoded by the cp32 plasmids, as suggested by their in vitro binding activities, our in vivo data along with previous in vivo analyses have demonstrated little or no attenuation of spirochetes lacking various *erp* or *revA* genes (Table 1) (Alitalo et al, 2002; Antonara et al, 2007; Brissette et al, 2009b; Lin et al, 2012; Lin et al, 2015; Lin et al, 2020; Moriarty et al, 2012). Even our current study demonstrated differential gene expression of four genes encoding surface-exposed proteins between wt and Δcp32 spirochetes in culture, whereas transcript levels for these genes were similar in both strains in vivo (*bbh06*, *bba24*, *bba25*, and *bbk32*) (Fig. 3). This further demonstrates the differences in spirochete gene expression in vitro and in vivo, and the frailty of extrapolating between them. *B. burgdorferi* outer surface proteins are reported to have multi-functional and redundant roles in vivo (Caine and Coburn, 2016), hence the utilization of *B. burgdorferi* derivatives lacking numerous cp32 plasmids opens the possibility for deeper investigation into the in vivo function of specific cp32-encoded genes, which we plan to further explore.

The ability of *B. burgdorferi* lacking all cp32 plasmids to complete the experimental mouse-tick infectious cycle raises the question as to why these plasmids are conserved in all sequenced Lyme disease *Borrelia* (Mongodin et al, 2013)? Perhaps the cp32 plasmids are not essential to complete the enzootic cycle in the lab environment but confer an unidentified selective advantage in nature. In fact, it has been noted that variant cp32 plasmids can be associated with different disease phenotypes and dissemination (Brisson et al, 2013; Lemieux et al, 2023; Purser and Norris, 2000). Additionally, as mixed strain infections in both the vertebrate host and the tick vector would more closely mimic the natural environment (Andersson et al, 2013; Brisson and Dykhuizen, 2004; Durand et al, 2017; Durand et al, 2015; Strandh and Råberg, 2015; Walter et al, 2016), such studies may shed light on cp32 maintenance in Lyme disease *Borrelia*. However, another rationale for the maintenance of these plasmids may be conferred by the endogenous prophage encoded by the cp32 plasmids (Eggers and Samuels, 1999). Further studies are needed to shed light on the conservation of the cp32 plasmids in Lyme disease *Borrelia*.

In conclusion, by engineering the first variant to lack all cp32 plasmids, we have shown that the Lyme disease spirochete does not require cp32 plasmids to complete the experimental mouse-tick infectious cycle. Competence of Δcp32 spirochetes throughout the mouse-tick infectious cycle enables future studies to define the role of cp32-encoded proteins, of phage during infection, and in

horizontal gene transfer. This study has defined the nonessential contribution of the cp32 plasmids to the infectious phenotype of the Lyme disease spirochete *Borrelia burgdorferi*.

## Methods

### Reagents and tools table

| Reagent/Resource | Reference or Source | Identifier or Catalog Number |
|---|---|---|
| **Experimental Models** | | |
| *E. coli* TOP10 | Invitrogen | C404006 |
| RML mice (*M. musculus*) | Outbred Swiss-Webster mice which have been reared as a closed colony at the Rocky Mountain Laboratories breeding facility since 1937 | N/A |
| C57BL/6 Strain 027 (*M. musculus*) | Charles River | C57BL/6NCrl Strain 027 |
| *B. burgdorferi* B31-A3-68-LS | Chu et al (2016); Gilbert et al (2007) | Infectious derivative of B31-A3-68 with *bbe02* on lp25 inactivated by the insertion of flgBp::*lacI* and flgBp::*aadA* (streptomycin resistance); contains all plasmids except cp9 and lp56. |
| *B. burgdorferi* B31-A3-68-LS Δcp32-3 | Wachter et al (2023) | Clonal derivative of B31-A3-68 LS. Missing plasmids cp9 and lp56, and cp32-3 plasmids. |
| *B. burgdorferi* B31-A3-68-LS Δcp32-1/3 | Wachter et al (2023) | Clonal derivative of B31-A3-68 LS. Missing plasmids cp9 and lp56, cp32-1, and cp32-3 plasmids. |
| *B. burgdorferi* B31-A3-68-LS Δcp32-1/3/4 | Wachter et al (2023) | Clonal derivative of B31-A3-68 LS. Missing plasmids cp9 and lp56, cp32-1, cp32-3, and cp32-4 plasmids. |
| *B. burgdorferi* B31-A3-68-LS Δcp32-1/3/4/6 | Wachter et al (2023) | Clonal derivative of B31-A3-68 LS. Missing plasmids cp9 and lp56, cp32-1, cp32-3, cp32-4, and cp32-6 plasmids. |
| *B. burgdorferi* B31-A3-68-LS Δcp32-1/3/4/6/7 | Wachter et al (2023) | Clonal derivative of B31-A3-68 LS. Missing plasmids cp9 and lp56, cp32-1, cp32-3, cp32-4, cp32-6, and cp32-7 plasmids. |
| *B. burgdorferi* B31-A3-68-LS Δcp32-1/3/4/6/7/8 | Wachter et al (2023) | Clonal derivative of B31-A3-68 LS. Missing plasmids cp9 and lp56, cp32-1, cp32-3, cp32-4, cp32-6, cp32-7, and cp32-8 plasmids. |
| *B. burgdorferi* B31-A3-68-LS Δcp32-1/3/4/6/7/8/9 (Δcp32) | Wachter et al (2023) | Clonal derivative of B31-A3-68 LS. Missing plasmids cp9 and lp56, cp32-1, cp32-3, cp32-4, cp32-6, cp32-7, cp32-8, and cp32-9 plasmids. |
| **Recombinant DNA** | | |
| pOG | Wachter et al (2023) | pBSV2G vector lacking cp9 genes |
| pOG::bbs31-35 | Wachter et al (2023) | pOG vector containing cp32-3 plasmid replication genes bbs31-35 |
| pOG::bbp29-32 | Wachter et al (2023) | pOG vector containing cp32-1 plasmid replication genes bbp29-32 |
| pOG::bbr29-33 | Wachter et al (2023) | pOG vector containing cp32-4 plasmid replication genes bbr29-33 |
| pOG::bbm29-32 | Wachter et al (2023) | pOG vector containing cp32-6 plasmid replication genes bbm29-32 |
| pOG::bbo29-32 | Wachter et al (2023) | pOG vector containing cp32-7 plasmid replication genes bbo29-32 |

| Reagent/Resource | Reference or Source | Identifier or Catalog Number |
|---|---|---|
| pOG::bbl29-32 | Wachter et al (2023) | pOG vector containing cp32-8 plasmid replication genes bbl29-32 |
| pOG::bbn29-32 | Wachter et al (2023) | pOG vector containing cp32-9 plasmid replication genes bbn29-32 |
| **Antibodies** | | |
| **Oligonucleotides and other sequence-based reagents** | | |
| L29-32.sacl.For | Wachter et al (2023) | GAA GAG CTC CAA CGG TTC CTA ATA GCT ATT AGC |
| M29-32.sacl.For | Wachter et al (2023) | GAA GAG CTC GGT GGT TCC TAA TAG CTA ACA GC |
| O29-32.sacl.For | Wachter et al (2023) | GAA GAG CTC GCT ACC TGC AAT TAA TCT AGC |
| P29-32.sacl.For | Wachter et al (2023) | GAA GAG CTC GCA ATG GGT CCT AAT AGT TAA CAG C |
| N29-32.sacl.For | Wachter et al (2023) | GAA GAG CTC GCA ACT AGT GCA AGT AGT ACC TGC |
| R29-33.sacl.For | Wachter et al (2023) | GAA GAG CTC GCA TAG ACG GCT TTG AAA ATA CTG C |
| bbs32.sacl.For | Wachter et al (2023) | GGA GAG CTC GCT TAT TTT TGC TTG AAT ATT TTC TTG TGG |
| L32_Kpnl_r | Wachter et al (2023) | GGT GGT ACC TTA ATA TAT CTC AAA AAA TTT TGA TAA TGC TTC |
| M32_Kpnl_r | Wachter et al (2023) | GGC TGG TAC CTT ATT TTA ATT TTG CAT AAA AAT TCA TTA ATG |
| N32_Kpnl_r | Wachter et al (2023) | GGC TGG TAC CTT ATT TTA TTT TTA ACA TTT TAT CTA AAG ATT TTT TAT ATT C |
| O32_Kpnl_r | Wachter et al (2023) | GTT GGT ACC TTA CCT GGA CAT GAT TAT TGT TAT AAA TTT AC |
| P32_Kpnl_r | Wachter et al (2023) | CCT GGT ACC TCA TAT TTT CAA AAT AAA ATT ATT TAA AAC GTT TAC |
| R33_Kpnl_r | Wachter et al (2023) | CCG TGG TAC CTT AAA ACA TAT TGT TTA ATA TTT TTT TTA TTT CTT G |
| S35_Kpnl_r | Wachter et al (2023) | CCG TGG TAC CTT ATA TTT TTT TTA AAA AAA TTT CTA AAA TAT TTT CAT ATT C |
| Cp9.IR1.Bcll | Wachter et al (2023) | GCT GAT CAT TGA TTA ATT TTT TGT GGA TTA TTG TTT AAG TTT TC |
| Cp9.ORF3.SacI | Wachter et al (2023) | GTG AGC TCG ATT TTT AGT TCT TCA TAT TTC TTT AAA AGT TTT TTA AG |
| cp32-1_f | Wachter et al (2023) | CATTAAGATTGATGCCGTGGAA |
| flaB_f | Hughes et al (2008) | TCTTTTCTCTGGTGAGGGAGCT |
| flaB_r | Hughes et al (2008) | TCCTTCCTGTTGAACACCCTCT |
| nid_f | Jewett et al (2007b) | CACCCAGCTTCGGCTCAGTA |
| nid_r | Jewett et al (2007b) | TCCCCAGGCCATCGGT |
| Actin.lx.sc.ForQPCR | Li et al (2013) | GATCATGTTCGAGACCTTCA |
| Actin.lx.sc.RevQPCR | Li et al (2013) | CGATACCCGTGGTACGA |
| bb0347_F | This study | GTG GCC ATC CTC GCC AAA TAT AAT AG |
| bb0347_R | This study | GAT TTC ACC TGT TGT TTC TTT TAT TTT TGG CCT TCT G |
| bbk32_F1 | This study | GAA TCC CCT GGC TTA TTT GAT AAG GGA AAC TC |
| bbk32_R1 | This study | CTT TCT AGC AAT CTT ACC TTT ACC TTT CTT ATT CAT AGG C |
| bba68_qF | This study | CCA ATT GCT TTT AGC TCT GAA GCG ATA GTT TCC |
| bba68_qR | This study | CAC CAA TCC GGG GGA AAA CAC C |
| bbh06_qF | This study | CCC CCT CAA GTT CTA CAG CAG C |
| bbh06_qR | This study | GGA GAT ATG GTT AAT GAT AGG GAA ATA AAT TCA AGA AGC |
| bbg26_f | This study | GCT TGT GAC TTC GTC TCT AAG CTT AAA AAG C |
| bbg26_r | This study | CGA CTC AAA GAT ACA ATT GAA CTC TCA ATA CAA GC |
| bba24_f | This study | CAC CAC TAC TTC CAG TTT CTT TGA G |
| bba24_r | This study | CCA TTC ATA CTT GAA GCA AAA GTG C |
| bba25_f | This study | CTCTCCACCATTTTCAATATTTTGTTTTTCC |
| bba25_r | This study | CCT ATA AAC ACA GCT GAA AGA TTG CTT G |

| Reagent/Resource | Reference or Source | Identifier or Catalog Number |
|---|---|---|
| **Chemicals, Enzymes and other reagents** | | |
| CMRL 1066 w/o L-Glutamine (Powder) | United States Biological | C5900-01 |
| Bacto Neopeptone | ThermoFisher Scientific | 211681 |
| Probumin® Bovine Serum Albumin Universal Grade | Sigma-Aldrich | 810037 |
| Bacto TC Yeastolate | ThermoFisher Scientific | 255772 |
| HEPES sodium salt | Millipore Sigma | 391333 |
| D-(+)-Glucose – powder | Millipore Sigma | G7021 |
| Sodium Citrate Dihydrate | Fisher Scientific Corp | 6132-04-3 |
| Pyruvic acid sodium salt | Millipore Sigma | P5280 |
| N-acetyl-D-glucosamine | Millipore Sigma | A4106 |
| Sodium bicarbonate | Sigma | S5761 |
| 500 RB SERUM S YG (O/S)500 ml | Pel-Freez | 31123 |
| Streptomycin sulfate | GoldBio | S-150-100 |
| Gentamicin sulfate salt powder | Millipore Sigma | G1264-5G |
| Rifampicin powder | Millipore Sigma | R3501-5G |
| Amphotericin B solubilized - 500 mg | Millipore Sigma | A9528-500MG |
| Phosphomycin disodium salt | Millipore Sigma | P5396-5G |
| Potassium sulfate | Millipore Sigma | P0772 |
| DreamTaq DNA Polymerase | ThermoFisher Scientific | EP0712 |
| dNTP set (100 mM) | ThermoFisher Scientific | 10297018 |
| MilliporeSigma™ GelRed™ Nucleic Acid Stain 10000X Water | FisherScientific | SCT123 |
| Trizol Reagent | ThermoFisher Scientific | 15596018 |
| RNAaseZap | ThermoFisher Scientific | AM9782 |
| DNase 1, RNase-free (1U/ul) | ThermoFisher Scientific | EN0521 |
| IQTM SYBR® Green Supermix | Bio-Rad | 1708880 |
| **Software** | | |
| GraphPad Prism 9.4.1 | https://www.graphpad.com/ | |
| BioRender | https://www.biorender.com/ | |
| **Other** | | |
| Quick-DNA/RNA Miniprep Plus Kit | Zymo Research | D7003 |
| High capacity cDNA Reverse Transcription kit | ThermoFisher Scientific | 4368814 |
| ViiA7 Real-Time PCR System | ThermoFisher Scientific | |
| StepOnePlus Real-Time PCR System | ThermoFisher Scientific | |

## Ethics statement

All animal work was performed according to the guidelines of the National Institutes of Health, *Public Health Service Policy on Humane Care and Use of Laboratory Animals* (Health, 2015), and the United States Institute of Laboratory Animal Resources, National Research

**Table 1. Demonstrated and inferred functions of proteins encoded by cp32 genes and proposed functionally redundant non-cp32 genes in *Borrelia burgdorferi*.**

| cp32-encoded | | | Functionally redundant non-cp32 genes[a] | | | |
| --- | --- | --- | --- | --- | --- | --- |
| Gene (genetic element) | Protein | Function[b] | Gene (genetic element) | Protein | Function[b] | References |
| bbp38 (cp32-1) & bbl39 (cp32-8) | ErpA (BbCRASP-5, OspE-related) | In vitro: Prevent complement activation by binding to complement regulator Factor H. Facilitate spirochete dissemination through plasminogen binding. In vivo: Slight attenuation of ErpA mutants in mice. | bba68 (lp54) & bbh06 (lp28-3) | CspA (BbCRASP-1) and CspZ (BbCRASP-2) | In vitro: Prevent complement activation by binding to complement regulator Factor H. Binds fibronectin, plasminogen, and laminin. In vivo: Not required for mouse infectivity. | (Alitalo et al, 2002; Brissette et al, 2009b; Bykowski et al, 2007; Hallström et al, 2010; Hart et al, 2018b; Lin et al, 2012; Lin et al, 2020; Metts et al, 2003; Stevenson et al, 2002) |
| bbn38 (cp32-9) | ErpP (BbCRASP-3, OspE-related) | | | | | |
| bbs41 (cp32-4) | ErpG (OspF-related) | In vitro: Binds heparan sulfate. In vivo: Not required for mouse infectivity, but adheres to vascular endothelium and promotes localization to joints, heart, and bladder in mice. | bba24 and bba25 (lp54) | DbpA and DbpB | In vitro: Binds decorin and dermatan sulfate. In vivo: Attenuation of mutants in mice. | (Antonara et al, 2007; Blevins et al, 2008; Fischer et al, 2003; Guo et al, 1998; Guo et al, 1995; Hyde et al, 2011; Imai et al, 2013; Lin et al, 2012; Lin et al, 2015; Parveen et al, 2003; Weening et al, 2008) |
| bbo39 (cp32-7) | ErpL (OspF-related) | | | | | |
| bbr42 (cp32-4) | ErpY (OspF-related) | | | | | |
| bbm38 (cp32-6) | ErpK (OspF-related) | | | | | |
| bbp39 (cp32-1) | ElpB (ErpB) | In vitro: Bind and inhibit the activity of the complement C1 complex. In vivo: function not determined. | bbk32 (lp36) | BBK32 | In vitro: Binds fibronectin and inactivates the activity of the C1 complement complex. In vivo: Not required for mouse infectivity, but mutants show mild attenuation in mice. | (Garcia et al, 2016; Hyde et al, 2011; Li et al, 2006; Lin et al, 2012; Moriarty et al, 2012; Norman et al, 2008; Pereira et al, 2022; Probert and Johnson, 1998; Seshu et al, 2006) |
| bbn39 (cp32-9) | ElpQ (ErpQ) | | | | | |
| bbm27 (cp32-6) & bbp27 (cp32-1) | RevA | In vitro: Binds fibronectin. Attenuation of bbm27 mutants in mice. In vivo: No role in vascular interactions in mice. | bb0347 (chromo) | BB0347 | In vitro: Interacts with the CS1 heparin-binding domain of human fibronectin. In vivo: No role in vascular interactions in mice. | (Brissette et al, 2009a; Gaultney et al, 2013; Li et al, 2006; Lin et al, 2012; Moriarty et al, 2012) |

[a]Genes proposed to encode proteins of similar functions to cp32-encoded genes.
[b]Predicted and inferred function in vitro and in vivo.

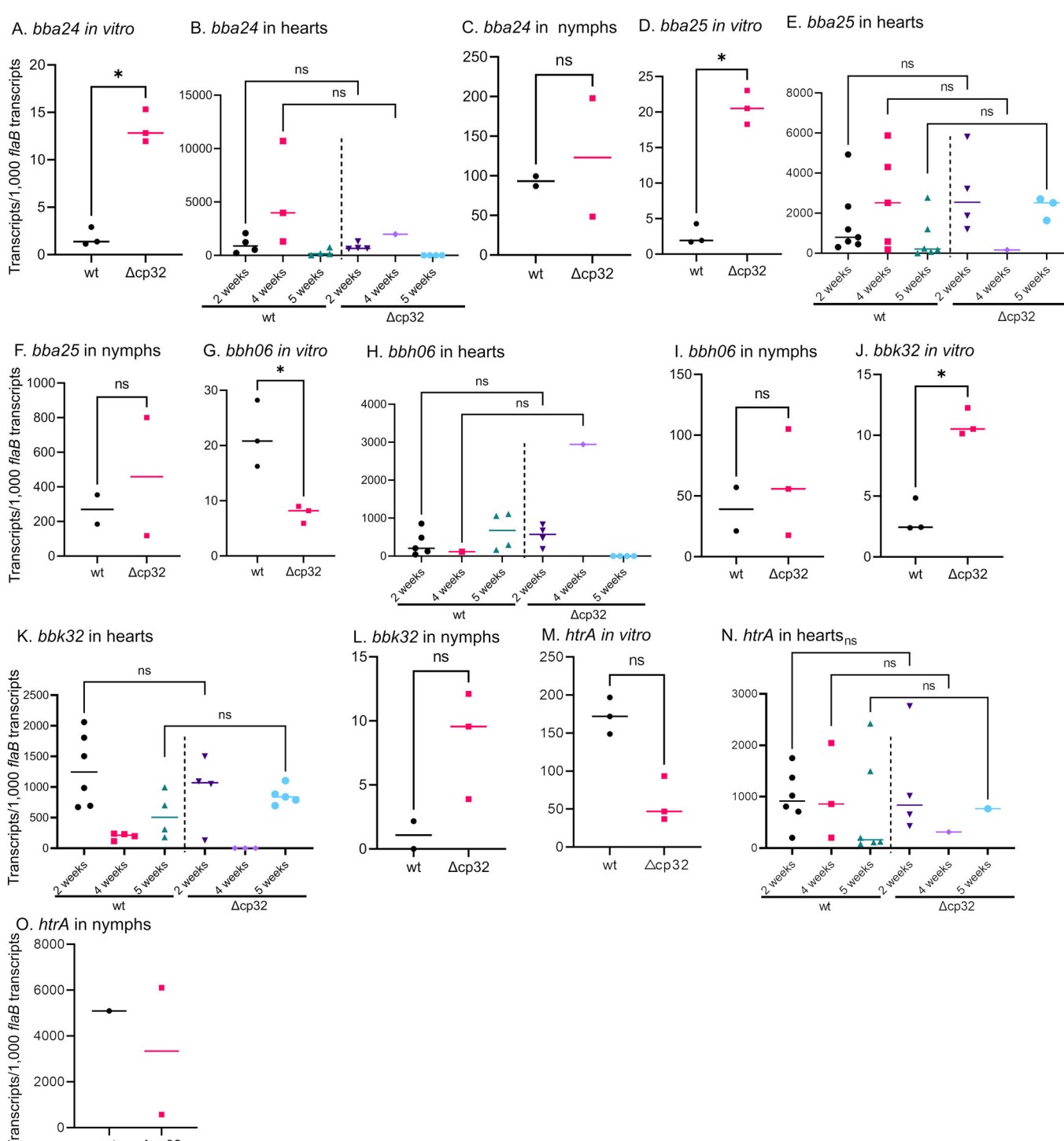

Council, *Guide for the Care and Use of Laboratory Animals* (Council, 2011) and the Canadian Council of Animal Care (CCAC, 2019). Protocols were approved by the Rocky Mountain Laboratories, NIAID, NIH Animal Care and Use Committee and the Animal Research Ethics Board at the University of Saskatchewan (AUP #20230079). The Rocky Mountain Laboratories are accredited by the International Association for Assessment and Accreditation of Laboratory Animal Care (AAALAC) and the University of Saskatchewan is a member of the University Animal Care Committee. All efforts to minimize animal suffering were made.

## B. burgdorferi strains and growth conditions

*B. burgdorferi* strains were cultured in Barbour-Stoenner-Kelly (BSK II) medium supplemented with 6% rabbit serum (PelFreez Biologicals, Rogers, AZ) and appropriate antibiotics (streptomycin,

◄ **Figure 3. Transcript levels of genes proposed to encode proteins of similar functions to cp32-encoded genes.**

(A–O) RNA was extracted from wt and Δcp32 cells in culture, from infected mouse hearts, and from replete infected nymphs. qRT-PCR was performed with primers specific for each gene target as described in Reagents and Tools Table. **(A)** *bba24* in vitro transcripts. **(B)** *bba24* transcripts from infected hearts. *bba24* could not be detected in hearts of C57BL/6 mice infected with Δcp32 at 5-weeks, but it was detected in RML mice at 5-weeks (data shown in source data but not graphed). **(C)** *bba24* transcripts from infected nymphs. **(D)** *bba25* in vitro transcripts. **(E)** *bba25* transcripts from infected hearts. **(F)** *bba25* transcripts from infected nymphs. **(G)** *bbh06* in vitro transcripts. **(H)** *bbh06* transcripts from infected hearts. *bbh06* could not be detected in hearts of C57BL/6 mice infected with Δcp32 at 5-weeks, but it was detected in RML mice at 5-weeks (data shown in source data but not graphed). **(I)** *bbh06* transcripts from infected nymphs. **(J)** *bbk32* in vitro transcripts. **(K)** *bbk32* transcripts from infected hearts. *bbk32* could not be detected in hearts of C57BL/6 mice infected with Δcp32 at 4-weeks. **(L)** *bbk32* transcripts from infected nymphs. **(M)** *htrA* in vitro transcripts. **(N)** *htrA* transcripts from infected hearts. **(O)** *htrA* transcripts from infected nymphs. Transcripts levels for all genes analyzed differed significantly between in vitro-grown Δcp32 and wt spirochetes (A, B, G, J) whereas in vivo transcript levels in infected mouse hearts and replete nymphs showed no significant difference between wt and Δcp32 (B, C, E, F, H, I, K, L, N, O). Data information: Data are presented in graphs as individual values of biological replicates, with the mean value indicated. The Mann–Whitney nonparametric test or the Dunn's multiple comparison of the Kruskal–Wallis test was used to determine significance of all qRT-PCR data. (A, D, G, J, M) $n = 3$ biological replicates. (B, E, H, K, N) $n = 7$ biological replicates with 1–7 yielding useable data. (C, F, I, L, O) $n = 2$–3 biological replicates. *$p$-value $< 0.05$ Source data are available online for this figure.

50 μg/mL; gentamicin, 40 μg/mL) at 35 °C under 2.5% $CO_2$ (Barbour, 1984). *B. burgdorferi* strain B31-A3-68-LS, which lacks linear plasmid lp56 and has *flgBp-lacI* inserted in the putative restriction-modification gene *bbe02* (Chu et al, 2016; Gilbert et al, 2007), was the wild-type strain used in this study. Cloning vectors were propagated using *E. coli* strain TOP10 (Invitrogen, Carlsbad, CA). All *B. burgdorferi* strains and derivatives, along with plasmids utilized in this study, are described in Reagents and Tools Table: Experimental Models.

## Assembly of constructs and transformation of *B. burgdorferi*

*B. burgdorferi* was transformed by electroporation as previously described (Samuels et al, 1994). Competent *B. burgdorferi* were freshly prepared from exponential phase culture and electroporated with 15–30 μg of plasmid DNA. Transformants were confirmed through PCR analysis and sequencing, and plasmid content was determined (Norris et al, 2011).

All primers used in this study are listed in Reagents and Tools Table: Oligonucleotides. The cp32 plasmids were systematically and sequentially displaced as previously described (Eggers et al, 2002; Stewart et al, 2001; Wachter et al, 2023). Briefly, a fragment encompassing the cp32 plasmid partition and replication genes from PFAM57 to PFAM32 for each cp32 of strain B31 was cloned into the multiple cloning site of the pOG shuttle vector (pBSV2G shuttle vector lacking the cp9 plasmid partition and replication genes) (Dulebohn et al, 2011; Jewett et al, 2007a). This resulted in an unstable shuttle vector that could displace the native cp32 plasmid, but was subsequently lost following three passages in the absence of antibiotic selection. Although the SV inserts included the *nucP*/pfam-161 ortholog of each B31 cp32 plasmid, this gene is not required for autonomous replication, incompatibility, or stability of the shuttle vector in *B. burgdorferi* ((Eggers et al, 2002; Stewart et al, 2001) P Rosa unpublished data). Prior to further studies, each newly obtained derivative was analyzed for plasmid content by multiplex PCR to ensure loss of the desired plasmid and retention of all other plasmids (see Fig. 1) (Bunikis et al, 2011).

## Quantitative PCR (qPCR) and quantitative reverse-transcriptase (qRT) PCR

Total RNA was extracted from in vitro grown *B. burgdorferi* at mid-log phase (5–7 × $10^7$ spirochetes/mL) using TRIzol reagent

(Life Technologies, Carlsbad, CA) per the manufacturer's instructions and treated with 1 unit of DNAse I (Ambion, Foster City, CA) for 1 h at 37 °C. Total RNA was extracted from murine hearts, joints and ticks with the *Quick*-DNA/RNA Miniprep Plus Kit (Zymo Research, Irvine, CA) per the manufacturer's instructions. RNA was then quantified and subjected to Agilent Bioanalyzer 2200 Tape Station (Agilent, Santa Clara, CA) quality assessment. Only those RNAs with RIN values ≥7.4 were retained for subsequent analyses. cDNA was generated from 1–3 μg of RNA with the High-Capacity cDNA Reverse Transcriptase kit (Life Technologies), per the manufacturer's instructions. Genomic DNA was extracted from mouse tissues (hearts, ears, and joints) through mechanical disruption as previously described (Morrison et al, 1999) or with the *Quick*-DNA/RNA Miniprep Plus Kit (Zymo Research).

qPCR and qRT-PCR reactions were performed using IQ™ SYBR® Green Supermix (Bio-Rad Life Sciences, Hercules, CA) with gene-specific primer sets (500 nM) (Reagents and Tools Table: Oligonucleotides) (Hughes et al, 2008; Jewett et al, 2007b; Li et al, 2013). Reactions were performed on a ViiA7 Real-Time PCR System and on a StepOne Plus Real-Time PCR System (Applied Biosystems, Foster City, CA) and analyzed with PRISM software (PRISM). Negative control reactions with primers lacking a template and using RNA samples that underwent cDNA reactions in the absence of reverse transcriptase were performed with each reaction to ensure Ct values did not result from primer-primer interactions or from contaminating genomic DNA. Additionally, the melt-curves were analyzed for all reactions.

## Experimental mouse-tick infection studies

Mouse infections were performed with 6- to 8-week-old female mice, consisting of RML mice, a derivative of outbred Swiss-Webster mice which have been reared as a closed colony at the Rocky Mountain Laboratories breeding facility since 1937, and C57BL/6 Strain 027 (Charles River). A total of 5–8 RML mice per *B. burgdorferi* derivative and 7–8 C57BL/6 mice per *B. burgdorferi* derivative per time point were utilized. Mice were housed at ambient humidity of 50% + 10%, at ambient temperature between 20.6 and 23.9 °C, and under a 12 h ON/12 h OFF light cycle. The omission of male mice has no effect on this study as both sexes are equally susceptible to *B. burgdorferi* infections. All mice were monitored daily by trained veterinary staff, and any abnormalities were reported and further monitored. Mice were co-housed unless

undergoing a larval or nymph feeding, during which time they were housed singly to prevent tick removal by cage mates. Once all ticks had fed to repletion, mice were again co-housed. At the set endpoint, all mice were euthanized by cervical dislocation under isoflurane anesthesia.

Prior to injection of mice, the plasmid profiles of the inoculated *B. burgdorferi* strains were determined to ensure retention of plasmids lp25, lp28-1, and lp36, which are required for infectivity (Jewett et al, 2007b; Norris et al, 2011; Purser and Norris, 2000). Mice were inoculated intraperitoneally ($4 \times 10^3$ spirochetes) and subcutaneously ($1 \times 10^3$ spirochetes) for a total inoculum of $5 \times 10^3$ spirochetes per mouse. Experiments on C57BL/6 mice were blinded from researchers until after the data had been obtained. The number of injected spirochetes determined by counting with Petroff-Hausser counting chambers and confirmed by plating aliquots of the inocula for colony forming units (CFU). For mice that were fed on by tick larva, mouse infection was assessed at 3 weeks post-inoculation by immunoblot analysis to assess seroconversion to *B. burgdorferi* antigens. To determine infection and spirochete load, tissues were harvested at 2-, 4-, and 5-weeks (following larval feeding) post-injection. Spirochetes were isolated from mouse tissues (ear, bladder, joint, and fat) through liquid culture, and DNA and RNA were extracted from mouse tissues (ear, heart, and joint).

Larval *Ixodes scapularis* were obtained from egg masses laid by engorged female ticks purchased from Oklahoma State University. When not feeding, *I. scapularis* were maintained at room temperature in bell jars over potassium sulfate-saturated water. To assess tick acquisition, approximately 100–200 naive *I. scapularis* larvae were fed to repletion on each infected mouse. Acquisition and retention of *B. burgdorferi* by larval ticks were assessed 1 week, 7 weeks and approximately 16 weeks (once fully molted into nymphs) after drop-off, and spirochete loads in a subset of ticks were determined through mechanical disruption and plating for CFU. Naive mice were fed upon by approximately 20 *I. scapularis* nymphs per each infected cohort. The number of *B. burgdorferi* in nymphs was assessed prior to feeding, during feeding, at repletion, 24 h, 48 h, 10 days, and 1 month after drop-off through mechanical disruption and plating for CFU. Mouse infectivity was checked 5 weeks following tick feeding by attempted isolation of spirochetes from tissues (ear, bladder, and fat) in BSK-II medium.

### Graphics

BioRender.com was used to create the images in Fig. 1A and the graphical abstract. GraphPad Prism version 9.4.1 for Windows (GraphPad Software, Boston, MA, USA) was used to create the graphs in Fig. 1C, Fig. 2A–D, Fig. 3A–0, and Fig EV2.

## Data availability

No primary datasets have been generated and deposited.

The source data of this paper are collected in the following database record: biostudies:S-SCDT-10_1038-S44319-025-00378-9.

## Peer review information

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

## Acknowledgements

We thank Maarten Voordouw, Aaron White, and Sazzhad Mahmood for their careful reviews and helpful comments on this manuscript. We also thank the VIDO Veterinary Services Branch which includes Carla Norleen, Rob Stevens, and Sherri Tetland for help with the animal studies. This research was supported by the Intramural Research Program of the National Institute of Allergy and Infectious Diseases, National Institutes of Health and by VIDO who receives operational funding from the Government of Saskatchewan through Innovation Saskatchewan and the Ministry of Agriculture and from the Canada Foundation for Innovation.

## Author contributions

**Chad Hillman**: Investigation; Methodology. **Hannah Theriault**: Investigation. **Anton Dmitriev**: Investigation. **Satyender Hansra**: Investigation. **Patricia A Rosa**: Conceptualization; Resources; Supervision; Funding acquisition; Investigation; Methodology; Project administration; Writing—review and editing; Co-last author sharing senior authorship. **Jenny Wachter**: Conceptualization; Data curation; Formal analysis; Supervision; Investigation; Visualization; Methodology; Writing—original draft; Project administration; Writing—review and editing; Co-last author sharing senior authorship.

Source data underlying figure panels in this paper may have individual authorship assigned. Where available, figure panel/source data authorship is listed in the following database record: biostudies:S-SCDT-10_1038-S44319-025-00378-9.

## Disclosure and competing interests statement

The authors declare no competing interests.

# Expanded View Figures

## A. Gel 1

## B. Gel 2

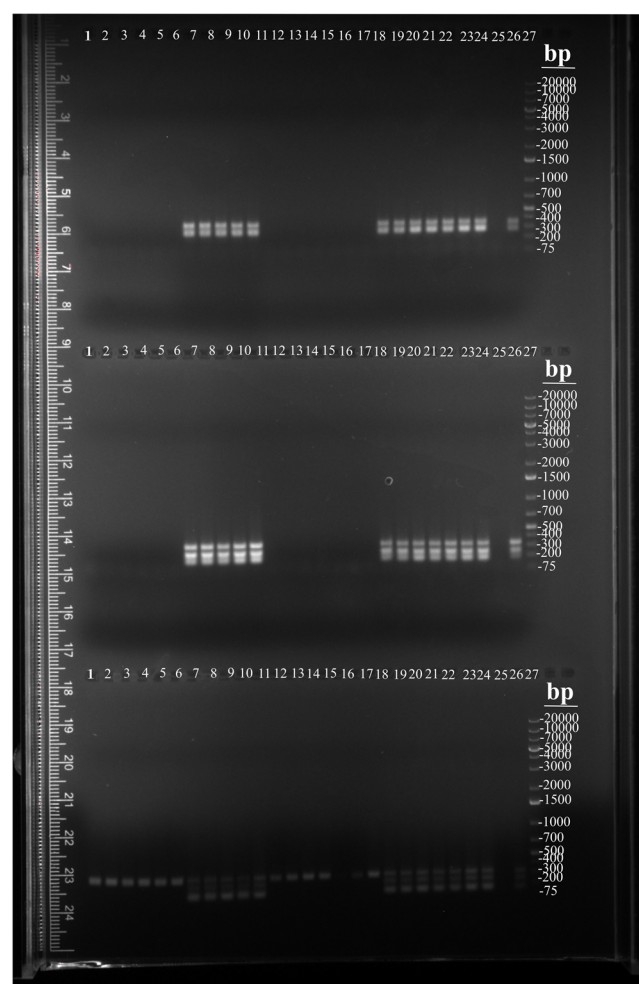

**Figure EV1. Multiplex PCR of murine ear outgrowths.**

To ensure that mice were infected with the correct *B. burgdorferi* derivative, outgrowths from ear tissues were analyzed by multiplex PCR using the circular plasmid primer sets. (**A**) Gel 1 was loaded as follows: 1. mouse 23-B (wt, 5 weeks); 2. mouse 134-N (wt, 4 weeks); 3. mouse 23-L (uninfected, 5 weeks); 4. mouse 23-R (uninfected, 5 weeks); 5. mouse 24-B (Δcp32, 5 weeks); 6. mouse 24-L (Δcp32, 5 weeks); 7. mouse 24-R (Δcp32, 5 weeks); 8. mouse 25-B (Δcp32, 5 weeks); 9. mouse 25-L (Δcp32, 5 weeks); 10. mouse 25-R (Δcp32, 5 weeks); 11. mouse 25-2 R (Δcp32, 5 weeks); 12. mouse 130-B (wt, 2 weeks); 13. mouse 130-L (wt, 2 weeks); 14. mouse 130-N (wt, 2 weeks); 15. mouse 130-R (wt, 2 weeks); 16. mouse 131-L (wt, 2 weeks); 17. mouse 131-N (wt, 2 weeks); 18. mouse 131-R (wt, 2 weeks); 19. mouse 132-L (Δcp32, 2 weeks); 20. Negative PCR control (water); 21. Positive PCR control (B31-A3); 22. Ladder (GeneRuler 1 kb plus). (**B**) Gel 2 was loaded as follows: 1. mouse 18-L (Δcp32, 2 weeks); 2. mouse 18-N (Δcp32, 2 weeks); 3. mouse 18-R (Δcp32, 2 weeks); 4. mouse 19-B (Δcp32, 2 weeks); 5. mouse 19-L (Δcp32, 2 weeks); 6. mouse 19-N (Δcp32, 2 weeks); 7. mouse 133-L (wt, 4 weeks); 8. mouse 133-N (wt, 4 weeks); 9. mouse 133-R (wt, 4 weeks); 10. mouse 134-L (wt, 4 weeks); 11. mouse 134-R (wt, 4 weeks); 12. mouse 21-L (Δcp32, 4 weeks); 13. mouse 21-N (Δcp32, 4 weeks); 14. mouse 21-R (Δcp32, 4 weeks); 15. mouse 22-B (Δcp32, 4 weeks); 16. mouse 22-L (Δcp32, 4 weeks); 17. mouse 22-N (Δcp32, 4 weeks); 18. mouse 22-R (Δcp32, 4 weeks); 19. mouse 136-B (wt, 5 weeks); 20. mouse 136-N (wt, 5 weeks); 21. mouse 136-R (wt, 5 weeks); 22. mouse 137-L (wt, 5 weeks); 23. mouse 137-N (wt, 5 weeks); 24. mouse 137-R (wt, 5 weeks); 25. Negative PCR control (water); 26. Positive PCR control (B31-A3); and 27. Ladder (GeneRuler 1 kb plus). Both (**A**) and (**B**) were loaded so that primer mix cp1 (amplifying cp32-8 at 375 bp, cp32-4 at 300 bp, and cp9 at 226 bp), cp2 (amplifying cp32-1 at 350 bp, cp32-6 at 276 bp, and cp32-7 at 200 bp), and cp3 (amplifying cp26 at 325 bp, cp32-9 at 250 bp, and cp32-3 at 170 bp) are represented by the top, middle, and bottom wells, respectively.

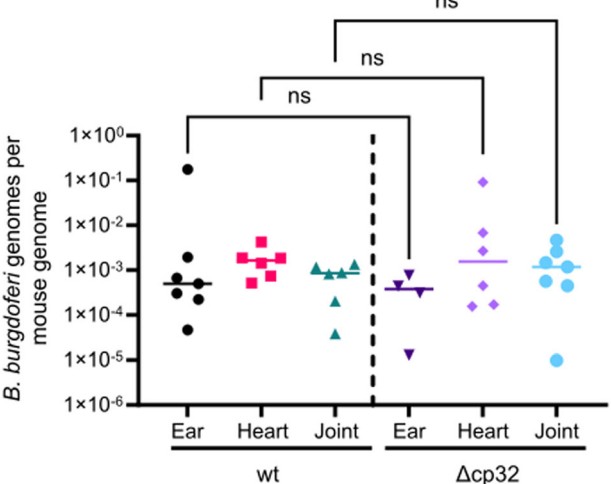

| Strain | Tissue | Number of tissues analyzed | Time point of tissue collection |
|--------|--------|----------------------------|----------------------------------|
| wt | Ear | 6 | 4 weeks |
| wt | Heart | 6 | 4 weeks |
| wt | Joint | 4 | 4 weeks |
| Δcp32 | Ear | 4 | 4 weeks |
| Δcp32 | Heart | 6 | 4 weeks |
| Δcp32 | Joint | 7 | 4 weeks |

**Figure EV2. Infectivity of wt and Δcp32 spirochetes in mice at 4-weeks post injection.**

Similar numbers of wt and Δcp32 genomes were detected by qPCR in ear, heart, and joint tissues of needle inoculated mice at 4-weeks post-injection. The number of tissues that were analyzed and the time points of tissue collection are shown as a table beneath the graph. Mann–Whitney nonparametric test was used to determine significance. Data are presented in graphs as individual values of biological replicates, with the mean value indicated. ***$p$-value < 0.0005. Source data are available online for this figure.

