## [Peer Review File · EMBO Reports]

Borrelia burgdorferi lacking all cp32 prophage plasmids retains full infectivity in mice

Chad Hillman, Hannah Theriault, Anton Dmitriev, Satyender Hansra, Patricia Rosa, and Jenny Wachter

Corresponding author(s): Jenny Wachter (jenny.wachter@usask.ca)

Review Timeline:

Submission Date:	4th Apr 23
Editorial Decision:	5th May 23
Revision Received:	4th Jul 23
Editorial Decision:	11th Aug 23
Revision Received:	29th Nov 24
Editorial Decision:	18th Dec 24
Revision Received:	9th Jan 25
Accepted:	17th Jan 25

Editor: Achim Breiling

Transaction Report:

Dear Dr. Wachter,

Thank you for the transfer of your manuscript to EMBO reports. I have now received the reports from the three referees that were asked to evaluate your study, which can be found at the end of this message.

As you will see, the referees state that these findings are of interest. However, they have several comments, concerns, and suggestions, indicating that a major revision of the manuscript is necessary to allow publication of the study in EMBO reports. As the reports are below, and all the referee concerns need to be addressed, I will not detail them here.

Given the constructive referee comments, I would like to invite you to revise your manuscript with the understanding that all referee concerns must be addressed in the revised manuscript and in a detailed point-by-point response. Acceptance of your manuscript will depend on a positive outcome of a second round of review. It is EMBO reports policy to allow a single round of revision only and acceptance of the manuscript will therefore depend on the completeness of your responses included in the next, final version of the manuscript.

1) a .docx formatted version of the final manuscript text (including legends for main figures, EV figures and tables), but without the figures included. Figure legends should be compiled at the end of the manuscript text.

Unless you plan to add substantial additional data, we would plan to publish your manuscript in the Report format. For a Scientific Report we require that results and discussion sections are combined in a single chapter called "Results & Discussion". Please do this for your revised manuscript. For more details, please refer to our guide to authors: <http://www.embopress.org/page/journal/14693178/authorguide#researcharticleguide>

2) individual production quality figure files as .eps, .tif, .jpg (one file per figure), of main figures (up to 8) and EV figures. Please upload these as separate, individual files upon re-submission.

For more details, please refer to our guide to authors: <http://www.embopress.org/page/journal/14693178/authorguide#manuscriptpreparation>

Please consult our guide for figure preparation: http://wol-prod-cdn.literatumonline.com/pb-assets/embo-site/EMBOPress_Figure_Guidelines_061115-1561436025777.pdf

See also the guidelines for figure legend preparation: <https://www.embopress.org/page/journal/14693178/authorguide#figureformat>

4) a complete author checklist, which you can download from our author guidelines (<https://www.embopress.org/page/journal/14693178/authorguide>). Please insert page numbers in the checklist to indicate where

the requested information can be found in the manuscript. The completed author checklist will also be part of the RPF.

5) that primary datasets produced in this study (e.g. RNA-seq, ChIP-seq, structural and array data) are deposited in an appropriate public database. If no primary datasets have been deposited, please also state this in a dedicated section (e.g. 'No primary datasets have been generated and deposited'), see below.

The accession numbers and database should be listed in a formal "Data Availability" section (placed after Materials & Methods) that follows the model below. This is now mandatory (like the COI statement). Please note that the Data Availability Section is restricted to new primary data that are part of this study. This section is mandatory. As indicated above, if no primary datasets have been deposited, please state this in this section

Data availability

8) Regarding data quantification and statistics, please make sure that the number "n" for how many independent experiments were performed, their nature (biological versus technical replicates), the bars and error bars (e.g. SEM, SD) and the test used to calculate p-values is indicated in the respective figure legends (also for potential EV figures and all those in the final Appendix). Please also check that all the p-values are explained in the legend, and that these fit to those shown in the figure. Please provide statistical testing where applicable. Please avoid the phrase 'independent experiment', but clearly state if these were biological or technical replicates. Please also indicate (e.g. with n.s.) if testing was performed, but the differences are not significant. In case n=2, please show the data as separate datapoints without error bars and statistics. See also: <http://www.embopress.org/page/journal/14693178/authorguide#statisticalanalysis>

9) Please also note our reference format:

10) We updated our journal's competing interests policy in January 2022 and request authors to consider both actual and perceived competing interests. Please review the policy <https://www.embopress.org/competing-interests> and update your competing interests if necessary. Please name this section 'Disclosure and Competing Interests Statement' and put it after the Acknowledgements section.

11) We now use CRediT to specify the contributions of each author in the journal submission system. CRediT replaces the author contribution section. Please use the free text box to provide more detailed descriptions and do not provide your final manuscript text file with an author contributions section. See also guide to authors: <https://www.embopress.org/page/journal/14693178/authorguide#authorshipguidelines>

I look forward to seeing a revised version of your manuscript when it is ready. Please let me know if you have questions or comments regarding the revision.

Yours sincerely,

Referee #1:

In this study, Wachter et al. generated a *B. burgdorferi* strain that lacks all cp32 plasmids. They compared the phenotypes of this strain with the parental wild-type strain during tick-transmitted or needle-inoculated *B. burgdorferi* infection in mice. The authors found no statistically significant different readouts between the cp32-lacking strain and the parental wild-type strain, concluding Unaltered infectious phenotype of the Lyme disease spirochete *Borrelia burgdorferi* lacking all cp32 prophage plasmids. Lyme borreliae encode many cp32 plasmids, which encode multiple sequentially similar genes (e.g., OspE, OspF, Elp, RevA, etc). Although the functions of those genes have been largely characterized in vitro, The roles of those genes in Lyme borreliae enzootic cycle are unclear because knocking out all these genes in an infectious *B. burgdorferi* background is very challenging. Therefore, the major strength of this paper is that this is one of the first few studies that attempt to determine the infectious phenotypes of cp32 in the Lyme borreliae enzootic cycle using a strain that lacks all cp32 plasmids. Although this strain has been generated and described in the authors' previous publication (PMID: 36639656), the implications of the results would provide the great impact in elucidating the roles of those cp32-encoded genes. However, the weakness is how the in vivo experiments were performed and analyzed in this work (see the comments below), which may change the conclusion of this work and the title of this manuscript.

1. The major strength of this work is that this is one of the first time that the role of cp32 in Lyme borreliae enzootic cycle has been systemically tested. Although the in vitro functions of the cp32-encoded genes have been examined and reported, there was only one study examining one of the cp32-encoded genes (PMID: 26150536). The hurdle of testing the role of cp32-encoded genes and cp32 plasmids is the functional redundancy from multiple copies of those genes and plasmids. This work overcame such a hurdle, which would potentially provide the implications of the roles of cp32 plasmids and all cp32-encoded genes in Lyme borreliae enzootic cycle. I think that this strength is very important but hidden in the manuscript. This may need to be emphasized.

2. (Statistical power and the number of replicates) In Fig. 2B and C, the spirochete burdens were compared at heart and ears from the mice infected with the cp32-lacking strain or the parental strain, and no statistical difference was observed. Unless I misread the manuscript, in some of the groups, the tissues from only one or two mice were used to compare the spirochete burdens, raising the question of whether the statistical power is sufficient to differentiate the difference of the burdens between the infection groups. If I misread the manuscript, and the authors used more than one or two mice and tissues for this work to make conclusions, they would need to clarify this. Otherwise, it is better to increase the sample size to provide more convincing statistical analysis as these results are important, driving the conclusions and the title of this manuscript.

3. (the discrepancy of this work vs. PMID: 26150536 and the tissues collected for the analysis) *B. burgdorferi* B31-A3, the strain that the WT strain in this study is derived from, encodes two copies of revA, which are located in cp32-1 and cp32-6 (PMID: 26150536). The B31-A3 deficient of both revA copies when introduced into mice subcutaneously at the dose of 10³ or 10⁴ showed defects of colonization at hearts but no defects at joints compared to the WT strain (Fig. 2 of PMID: 26150536). Therefore, one would assume the statistically significant different burdens of WT vs. cp32-lacking strain at hearts in Fig. 2B of this paper if hearts from more mice were used. If the authors use more mice (see the comment 1) and observe difference or no difference in the burdens at hearts, the discussion of similarity and difference between this work and PMID: 26150536 may need to be discussed. Further, the defects observed in heart but not joints in PMID: 26150536 suggests multiple tissues may be needed to make the conclusion supporting the title of this work. It is great that the authors have shown two tissues (i.e., ears and heart) in Fig. 2B and C. If the authors happened to collect more tissues from the mice or are going to add more mice in each infection group (see comment 1), it would be great to show the results from those tissues to support the conclusions.

4.(The infection route and the dose of infection) The needle infection experiments are stated to be performed via intraperitoneal inoculation (4 x 10³ spirochetes) and subcutaneous inoculation (1 x 10³ spirochetes) (see page 8, the section "Experimental mouse-tick infection studies" of the manuscript). The only needle infection experiment listed in the paper appears to be the Fig. 2B, but it is not clear if those mice in Fig. 2B were inoculated intraperitoneally or subcutaneously. The intraperitoneal inoculation is less physiologically relevant for Lyme borreliae infection, compared to other routes, as other routes (i.e., subcutaneous or intradermal route) would be closer to what occurs during tick bites. Therefore, if the intraperitoneal route was indeed used in Fig. 2B, either a strong rationale may need to be provided or the work may need to be redone using subcutaneous or intradermal

route of infection.

Additionally, the ID50 of the subcutaneous route of the infection by the *B. burgdorferi* strain B31-A3, the strain that the WT strain of this study is derived from, is between 103 (PMID: 16714588) to 104 (PMID: 27550932). If the Fig. 2B was indeed performed using subcutaneous injection with the dose of 1×10^3 spirochetes, half or more than half of the mice infected with the WT strain would be uninfected. That may make the results difficult to analyze. Therefore, unless the WT strain used in this study displays significant lower ID50 than B31-A3, it may be better to use a higher dose for the subcutaneous injection for the experiment in Fig. 2B.

5. Lyme borreliae have the potential to miss plasmids or develop mutations in each generation. The fact that the generation of the cp32-lacking strain required extensive passaging (Fig. 1A), it is quite possible either plasmid missing or the mutations occur. Such mutations or plasmid missing may impact the expression of other genes, resulting in the difference observed in Fig. 3. It is great that the authors have confirmed that the plasmids essential for in vivo survival of *B. burgdorferi* remain. Would the authors be able to confirm that other plasmids also remain or loss? I know that understanding whether there are mutations in other genes could be out of scopes. However, it would be better to discuss the possibility and implications of the differences of expression for the genes listed in Fig. 3 in the discussion.

6. It may be better to label the time points of tissue collection in Fig. 2B and C. The X-axis of Fig. 2D would need to be labeled.

7. For Fig. 2 and 3, because of the number of samples differ between different groups, the number of replicates may need to be spelled out in the figure legends, which will help readers to get the implications of the results.

Referee #2:

Wachter and colleagues have reported on the construction of a *B. burgdorferi* clone lacking all cp32s. Based on the ubiquitous nature of cp32-like plasmids throughout the *Borrelia* genus, the number of different cp32s maintained within a single cell, and the expression of many cp32-encoded genes within the mammalian host environment, many researchers have hypothesized that these plasmids must have critical roles within the enzootic cycle. Unexpectedly, but convincingly, Wachter and colleagues have now demonstrated that a *B. burgdorferi* clone lacking all cp32s can complete an experimental infection cycle with no notable defect relative to a wild-type clone. They also demonstrated that genes encoding products predicted to have similar functions to those encoded on cp32s are not over-expressed during the completion of this cycle. This suggests that the loss of potential important functions encoded on cp32s are not being masked by redundant genes.

The manuscript is succinct, clearly written, and interesting. Although largely reporting a negative result that cp32s are not required for the successful completion of the experimental infectious cycle, this is a critical and important finding for researchers within the Lyme disease field. The collective role of the cp32s has been the subject of much investigation. Furthermore, the careful and technical methodology used to eliminate the cp32 plasmids from a *B. burgdorferi* strain may be of interest to anyone interested in displacing native plasmids in another model bacterial system. Finally, this work serves as a cautionary tale that genetic abundance in bacteria may not be well-correlated with obvious biological importance.

Specific comments (all minor)

In the 2nd paragraph, the prophage sequences integrated into the lp56 have done so in the middle of the putative late phage operon (as demonstrated by Zhang & Marconi 2005) and have two other point mutations within that same region (see Schwartz, 2021). Thus, it is likely that the cp32 integrated into lp56 is not a prophage, but a prophage-like element.

In Figure 1B, the cp32-1/3/4 had what looks to be significantly poorer growth than any of the other clones. Clearly, clones lacking even more cp32s grow nearly as well as the wild-type, but I wondered if the authors had any potential explanation for the reproducibly poorer growth of this clone? (I wonder if it hints at interplay between certain elements of different cp32s? The loss of the early ones may have resulted in some sort of overabundance or deleterious effect until the others were also eliminated?)

In Figure 1C, it looks like cp32-1/3/4/6/7 was not included in the growth analysis. Is there a reason for this?

Could the authors expand a little on the strain of mice used in this study? My understanding of Swiss-Webster mice is that they are an inbred colony. Is the same true of these derivative RML mice? Could that be contributing to the lack of phenotype observed with the delta-cp32 *B. burgdorferi* clone? Is it known how infection in the RML mice compares (in terms of titer for infectivity and numbers of spirochetes recovered) to *Peromyscus* mice (a natural reservoir for *B. burgdorferi* in the United States)? While this does not detract from the findings of the authors of this paper, perhaps in the discussion the authors could acknowledge that *B. burgdorferi* may find itself in many different hosts within the natural enzootic cycle and cp32s could play a role (perhaps not essential) in other types of hosts or in outbred populations of hosts.

The displacement of native plasmids in *Borrelia burgdorferi* using an introduced replicon was first demonstrated with the cp9 of N40 by Stewart, et al. (2001) and subsequently also with a cp32 by Eggers, et al. (2002). The latter was with a region virtually identical to the one used in this work. Perhaps one or both of these should be cited as proof of principle for this methodology.

Casjens, et al. (2011) published the sequences of several members of the *Borrelia burgdorferi* sensu lato cohort. One of the isolates, *Borrelia garinii* Far04--an isolate recovered from puffin blood--contained no cp32s. I believe this was the first report of a member of the Lyme disease spirochetes recovered without any of these plasmids. While the work presented here is much more robust and involves a clone of *B. burgdorferi* (not *B. garinii*), Wachter, et al. may wish to acknowledge this work with a native member of the Lyme disease spirochetes, as it strengthens the findings of that cp32s may not be essential for the completion of the enzootic cycle even in natural hosts.

In the methods section, Assembly of constructs and transformation of *B. burgdorferi*, the authors state that the nucP/Pfam-161 gene was included in their constructs and is not necessary for replication, stability, or incompatibility and then cite unpublished data. This was demonstrated and published by Eggers, et al. (2002), in which it was shown that only the Pfam-57 and the region upstream of that gene were required for replication and stable maintenance and did not result in incompatibility.

Referee #3:

In this manuscript, the authors investigated whether cp32 plasmids are required for the viability of *Borrelia* in both mice and ticks, and demonstrated that cp32 plasmids were dispensable for transmission of *Borrelia* between mice and ticks. This study is important as it gives additional evidence for redundancies among adhesive surface lipoproteins. In addition, generating the Δ cp32 strain involved tremendous effort and provides an invaluable tool for investigating the role of prophages and the function of cp32-encoded lipoproteins. However, critical experiments presented in this manuscript appear to be incomplete as described below. If obtaining more complete data is not feasible, I strongly recommend that the authors acknowledge and discuss each of these issues in the text.

Major points:

- 1 It is currently understood that *B. burgdorferi* outer surface proteins have multifaceted and redundant roles in complement component interactions as well as tissue targeted adhesion and colonization. This information seems critical to understanding the authors results and so should be featured prominently in the introduction.
- 2 The introduction should also carefully summarize the past results on cp32-encoded lipoproteins including how those studies were done, rather than making passing mention of those studies in the results. This could help identify potential sources of discrepancies and provide clearer context for the current observations. For instance, the transposon mutant strain background used in Li et al. also lacks lp28-4, which encodes a few factors implicated in virulence including lipoproteins and complement regulating factors (e.g. WP_010890330.1). This raise the possibility that the transposon mutants decreased infectivity could be a result of the absence of lipoproteins encoded on lp28-4 as well as one or more cp32.
- 3 The authors are encouraged to design and perform their experiments so that the outcomes are comparable to the past studies.
 - 3.A Li et al. identified transposon mutants of bbm38 and bbo39 (encoding ErpK and ErpM, respectively) be potentially important for infectivity. They examined five tissue types (ear, heart, joints, inoculation site (skin), and bladder) at two time points (2- and 4-week post-infection). The authors only examined two tissue types (ear and heart) at only one late time point (5-weeks post-infection).
 - 3.B In particular, the authors are urged to examine infectious phenotypes early postinfection since cp32 encoding gene products are suggested to be involved in innate immune evasion. Analyzing only 5-week specimens (after establishment of infection) could minimize or obscure any effect that takes place early.
- 4 The authors appear to focus on an overly restrictive hypothesis on how redundancies operate in cp32 encoded lipoproteins and also only examined transcript levels of a set of lipoproteins whose binding targets and functions are reported to be overlapping (Figure 3). It is possible that the regulation does not occur at the transcript level - for instance, they might occur through post-translational modifications. In addition, when the bacteria are challenged by the host immune response, redundant functions may not be due to the most similar proteins, but could be from factors functioning in parallel pathways facilitating innate immune evasion. Specifically, there are a number of adhesive surface lipoproteins reported in *Borrelia* including OspC, RevB, and VlsE. Investigating the hypothesized redundancy may require using a tool that has not currently been established (e.g. multiple target CRISPRi in vivo), but at least the authors should expand their discussion of how lipoprotein redundancy may influence innate immune evasion in *Borrelia*.
- 5 It is highly recommended for the number of the experimental animals used for wildtype and Δ cp32 to be more similar. In Figure 2B, there appears to be one data point for heart in Δ cp32. Similarly, there appears to be one data point for ear in wild-type in Figure 2C. Variable samples numbers, especially when so small, lowers the validity of the statistical comparisons.
- 6 In Figure 3, bb0347 in nymph and Imp1 in hearts were not analyzed. Imp1 is also missing from Table 1.

Minor points:

1 Figure 2D, life stage (2) Unfed nymph, there is a two orders of magnitude difference in mean between wild-type and Δ cp32 strains and there generally seems to be more spirochete burden in Δ cp32 than in wild-type unfed nymph. In the text, the authors stated "The spirochete burdens in replete larvae and unfed nymphs were also similar between *I. scapularis* larvae infected with wt and Δ cp32 *B. burgdorferi*" - this seems like it could be statistical insignificance due to low samples numbers and thus is at best weak evidence.

2 "The ability of *B. burgdorferi* lacking all cp32 plasmids to complete the experimental mouse-tick infectious cycle raises the question as to why these plasmids are conserved in all sequenced Lyme disease *Borrelia* (Mongodin et al., 2013)? Perhaps the cp32 plasmids are not essential to complete the enzootic cycle in the lab environment but confer an unidentified selective advantage in nature." - This argument can be strengthened by noting the association of cp32 variant plasmids with disease phenotypes. The recent genome sequencing of clinical non-clinical isolates of Lyme-causing *Borrelia* found a significant association of cp32-4 and cp32-11 with dissemination (Lemieux 2023 (doi: 10.1101/2023.02.26.530159)).

3 There is a typo on page 4, line 21. Serconversion -> Seroconversion.

4 The following statement from the discussion: "The in vivo data from this study, as well as published data cited in Table 1, demonstrate the shortcomings of using in vitro binding assays to infer the in vivo functions of genes." seems needlessly combative as it only shows the shortcomings of overinterpretation of in vitro data. Especially given the known redundancy of adhesive lipoproteins, no observed negative effect from absence of cp32 and no statistically significant differences in transcripts from a subset of the non-cp32 genes does not demonstrate that the interaction detected in in vitro studies does not occur in vivo.

Referee #1:

In this study, Wachter et al. generated a *B. burgdorferi* strain that lacks all cp32 plasmids. They compared the phenotypes of this strain with the parental wild-type strain during tick-transmitted or needle-inoculated *B. burgdorferi* infection in mice. The authors found no statistically significant different readouts between the cp32-lacking strain and the parental wild-type strain, concluding Unaltered infectious phenotype of the Lyme disease spirochete *Borrelia burgdorferi* lacking all cp32 prophage plasmids. Lyme borreliae encode many cp32 plasmids, which encode multiple sequentially similar genes (e.g., OspE, OspF, Elp, RevA, etc). Although the functions of those genes have been largely characterized in vitro, The roles of those genes in Lyme borreliae enzootic cycle are unclear because knocking out all these genes in an infectious *B. burgdorferi* background is very challenging. Therefore, the major strength of this paper is that this is one of the first few studies that attempt to determine the infectious phenotypes of cp32 in the Lyme borreliae enzootic cycle using a strain that lacks all cp32 plasmids. Although this strain has been generated and described in the authors' previous publication (PMID: 36639656), the implications of the results would provide the great impact in elucidating the roles of those cp32-encoded genes. However, the weakness is how the in vivo experiments were performed and analyzed in this work (see the comments below), which may change the conclusion of this work and the title of this manuscript.

We thank Referee #1 for their careful review of the manuscript. We are happy that Referee #1 identified our manuscripts major strength in defining that *B. burgdorferi* derivatives lacking all cp32 plasmids are able to survive the mouse-tick infectious cycle. As our previous manuscript did not describe the ability of this *B. burgdorferi* derivative to transit the experimental infectious cycle, we are excited to publicize these results. We appreciate the helpful, constructive comments and have addressed them in full below.

1. The major strength of this work is that this is one of the first time that the role of cp32 in Lyme borreliae enzootic cycle has been systemically tested. Although the in vitro functions of the cp32-encoded genes have been examined and reported, there was only one study examining one of the cp32-encoded genes (PMID: 26150536). The hurdle of testing the role of cp32-encoded genes and cp32 plasmids is the functional redundancy from multiple copies of those genes and plasmids. This work overcame such a hurdle, which would potentially provide the implications of the roles of cp32 plasmids and all cp32-encoded genes in Lyme borreliae enzootic cycle. I think that this strength is very important but hidden in the manuscript. This may need to be emphasized.

Thank you for this! We have tried to further emphasize this in the Introduction on lines 118-119.

2. (Statistical power and the number of replicates) In Fig. 2B and C, the spirochete burdens were compared at heart and ears from the mice infected with the cp32-lacking strain or the parental strain, and no statistical difference was observed. Unless I misread the manuscript, in some of the groups, the tissues from only one or two mice were used to compare the spirochete burdens, raising the question of whether the statistical power is sufficient to differentiate the difference of the burdens between the infection groups. If I misread the manuscript, and the authors used more than one or two mice and tissues for this work to make conclusions, they would need to clarify

this. Otherwise, it is better to increase the sample size to provide more convincing statistical analysis as these results are important, driving the conclusions and the title of this manuscript. Results lines 156-157 and 174-175. This is correct, unfortunately, we did not extract DNA from every mouse and not all DNA extractions were successful. Thus, we only had usable DNA for qPCR from one or two mice in some instances. As we do not have more infected mouse tissues at this time, this work is planned for future experiments, but will not be complete for this manuscript. We want this manuscript to reflect the fact that cp32-lacking *B. burgdorferi* are able to survive the experimental mouse-tick infectious cycle and have revised the manuscript to reflect this.

3. (the discrepancy of this work vs. PMID: 26150536 and the tissues collected for the analysis) *B. burgdorferi* B31-A3, the strain that the WT strain in this study is derived from, encodes two copies of revA, which are located in cp32-1 and cp32-6 (PMID: 26150536). The B31-A3 deficient of both revA copies when introduced into mice subcutaneously at the dose of 103 or 104 showed defects of colonization at hearts but no defects at joints compared to the WT strain (Fig. 2 of PMID: 26150536). Therefore, one would assume the statistically significant different burdens of WT vs. cp32-lacking strain at hearts in Fig. 2B of this paper if hearts from more mice were used. If the authors use more mice (see the comment 1) and observe difference or no difference in the burdens at hearts, the discussion of similarity and difference between this work and PMID: 26150536 may need to be discussed. Further, the defects observed in heart but not joints in PMID: 26150536 suggests multiple tissues may be needed to make the conclusion supporting the title of this work. It is great that the authors have shown two tissues (i.e., ears and heart) in Fig. 2B and C. If the authors happened to collect more tissues from the mice or are going to add more mice in each infection group (see comment 1), it would be great to show the results from those tissues to support the conclusions.

Thank you for this comment; we agree that more *in vivo* work needs to be performed with these strains, but believe our preliminary analyses of mouse tissues are consistent with the major conclusions and merit inclusion in this manuscript. While this work is planned, we will not be able to complete it for this manuscript. Despite this, our main objective is to show the ability of cp32-lacking *B. burgdorferi* to survive the experimental mouse-tick infectious cycle. Therefore, we have revised the manuscript to fulfill this objective.

4.(The infection route and the dose of infection) The needle infection experiments are stated to be performed via intraperitoneal inoculation (4 x 10³ spirochetes) and subcutaneous inoculation (1 x 10³ spirochetes) (see page 8, the section "Experimental mouse-tick infection studies" of the manuscript). The only needle infection experiment listed in the paper appears to be the Fig. 2B, but it is not clear if those mice in Fig. 2B were inoculated intraperitoneally or subcutaneously. The intraperitoneal inoculation is less physiologically relevant for Lyme borreliosis infection, compared to other routes, as other routes (i.e., subcutaneous or intradermal route) would be closer to what occurs during tick bites. Therefore, if the intraperitoneal route was indeed used in Fig. 2B, either a strong rationale may need to be provided or the work may need to be redone using subcutaneous or intradermal route of infection.

Additionally, the ID₅₀ of the subcutaneous route of the infection by the *B. burgdorferi* strain B31-A3, the strain that the WT strain of this study is derived from, is between 103 (PMID: 16714588) to 104 (PMID: 27550932). If the Fig. 2B was indeed performed using subcutaneous injection with the dose of 1 x 10³ spirochetes, half or more than half of the mice infected with

the WT strain would be uninfected. That may make the results difficult to analyze. Therefore, unless the WT strain used in this study displays significant lower ID50 than B31-A3, it may be better to use a higher dose for the subcutaneous injection for the experiment in Fig. 2B.

Sorry for the confusion; mice were injected by both a subcutaneous AND intraperitoneal injection, for a total inoculum of 5×10^3 spirochetes. This has been fixed in the Materials and Methods section on lines 374-375. We included the data from needle-inoculated mice to show that there was no difference in infection between needle-inoculated or tick-fed mice.

5. Lyme borreliæ have the potential to miss plasmids or develop mutations in each generation. The fact that the generation of the cp32-lacking strain required extensive passaging (Fig. 1A), it is quite possible either plasmid missing or the mutations occur. Such mutations or plasmid missing may impact the expression of other genes, resulting in the difference observed in Fig. 3. It is great that the authors have confirmed that the plasmids essential for in vivo survival of *B. burgdorferi* remain. Would the authors be able to confirm that other plasmids also remain or loss? I know that understanding whether there are mutations in other genes could be out of scopes. However, it would be better to discuss the possibility and implications of the differences of expression for the genes listed in Fig. 3 in the discussion.

We analyzed each transformant for the presence of plasmids throughout cloning. To ensure that we analyzed clonal derivatives that retained all plasmids other than the displaced plasmid, we performed multiplex PCR on outgrowths from individual colonies to confirm retention of all other plasmids. We added a sentence to indicate this in the Materials and Methods section on lines 337-339.

6. It may be better to label the time points of tissue collection in Fig. 2B and C. The X-axis of Fig. 2D would need to be labeled.

We have added a table for both Fig. 2B and C to state the time points of tissue collection and have labeled the x-axis of Fig. 2D.

7. For Fig. 2 and 3, because of the number of samples differ between different groups, the number of replicates may need to be spelled out in the figure legends, which will help readers to get the implications of the results.

We have added a table for Fig 2 to show the number of replicates for each tissue.

Referee #2:

Wachter and colleagues have reported on the construction of a *B. burgdorferi* clone lacking all cp32s. Based on the ubiquitous nature of cp32-like plasmids throughout the *Borrelia* genus, the number of different cp32s maintained within a single cell, and the expression of many cp32-encoded genes within the mammalian host environment, many researchers have hypothesized that these plasmids must have critical roles within the enzootic cycle. Unexpectedly, but convincingly, Wachter and colleagues have now demonstrated that a *B. burgdorferi* clone lacking all cp32s can complete an experimental infection cycle with no notable defect relative to a wild-type clone. They also demonstrated that genes encoding products predicted to have similar functions to those encoded on cp32s are not over-expressed during the completion of this

cycle. This suggests that the loss of potential important functions encoded on cp32s are not being masked by redundant genes.

The manuscript is succinct, clearly written, and interesting. Although largely reporting a negative result that cp32s are not required for the successful completion of the experimental infectious cycle, this is a critical and important finding for researchers within the Lyme disease field. The collective role of the cp32s has been the subject of much investigation. Furthermore, the careful and technical methodology used to eliminate the cp32 plasmids from a *B. burgdorferi* strain may be of interest to anyone interested in displacing native plasmids in another model bacterial system. Finally, this work serves as a cautionary tale that genetic abundance in bacteria may not be well-correlated with obvious biological importance.

We thank Referee #2 for their careful, positive review of our manuscript. We appreciate their helpful, constructive comments and have addressed them in full below.

Specific comments (all minor)

In the 2nd paragraph, the prophage sequences integrated into the lp56 have done so in the middle of the putative late phage operon (as demonstrated by Zhang & Marconi 2005) and have two other point mutations within that same region (see Schwartz, 2021). Thus, it is likely that the cp32 integrated into lp56 is not a prophage, but a prophage-like element.

Thank you for this! We have added this in the second paragraph on line 61 and have added the corresponding references at the end of the sentence on line 57.

In Figure 1B, the cp32-1/3/4 had what looks to be significantly poorer growth than any of the other clones. Clearly, clones lacking even more cp32s grow nearly as well as the wild-type, but I wondered if the authors had any potential explanation for the reproducibly poorer growth of this clone? (I wonder if it hints at interplay between certain elements of different cp32s? The loss of the early ones may have resulted in some sort of overabundance or deleterious effect until the others were also eliminated?)

Thank you for this comment. I checked the raw data on this strain and indeed, all three biological replicates failed to reach 10^8 by the end of the growth curve. I will re-analyze the growth of this derivative when I have a lot of BSA (essential component of undefined BSKII growth medium) that supports growth similar to the BSA used for the previous experiments, but it is currently unavailable.

In Figure 1C, it looks like cp32-1/3/4/6/7 was not included in the growth analysis. Is there a reason for this?

Unfortunately, I do not have the data for this strain. Dr. Rosa has retired and I have just started my lab. As described above, I been unable to acquire a source of BSA that will support the growth of *B. burgdorferi* comparable to previous experiments. I am in the process of acquiring more lots and I will start these cultures for growth curves once I have BSA that supports growth similar to the Sigma BSA used for the previous experiments. We do have data demonstrating that a Δ cp32-1/3/4/6/7 derivative in a different background (pflac::ibbd18) has a similar growth curve to wt cells. However, since this background is not described in the current manuscript, we have not used these data.

Could the authors expand a little on the strain of mice used in this study? My understanding of Swiss-Webster mice is that they are an inbred colony. Is the same true of these derivative RML mice? Could that be contributing to the lack of phenotype observed with the delta-cp32 B. burgdorferi clone? Is it known how infection in the RML mice compares (in terms of titer for infectivity and numbers of spirochetes recovered) to Peromyscus mice (a natural reservoir for B. burgdorferi in the United States)? While this does not detract from the findings of the authors of this paper, perhaps in the discussion the authors could acknowledge that B. burgdorferi may find itself in many different hosts within the natural enzootic cycle and cp32s could play a role (perhaps not essential) in other types of hosts or in outbred populations of hosts.

RML mice originated from outbred Swiss-Webster mice and have been bred as a closed colony since 1937. We have added this on lines 361-362 of the Materials and Methods, and mentioned Peromyscus as a natural reservoir host on lines 258-262 in the Discussion.

The displacement of native plasmids in Borrelia burgdorferi using an introduced replicon was first demonstrated with the cp9 of N40 by Stewart, et al. (2001) and subsequently also with a cp32 by Eggers, et al. (2002). The latter was with a region virtually identical to the one used in this work. Perhaps one or both of these should be cited as proof of principle for this methodology.

We have cited previous work by Stewart and Eggers on lines 328-329 of the Methods section.

Casjens, et al. (2011) published the sequences of several members of the Borrelia burgdorferi sensu lato cohort. One of the isolates, Borrelia garinii Far04--an isolate recovered from puffin blood--contained no cp32s. I believe this was the first report of a member of the Lyme disease spirochetes recovered without any of these plasmids. While the work presented here is much more robust and involves a clone of B. burgdorferi (not B. garinii), Wachter, et al. may wish to acknowledge this work with a native member of the Lyme disease spirochetes, as it strengthens the findings of that cp32s may not be essential for the completion of the enzootic cycle even in natural hosts.

A sentence about this was added with citation on lines 63-66 of the Introduction.

In the methods section, Assembly of constructs and transformation of B. burgdorferi, the authors state that the nucP/Pfam-161 gene was included in their constructs and is not necessary for replication, stability, or incompatibility and then cite unpublished data. This was demonstrated and published by Eggers, et al. (2002), in which it was shown that only the Pfam-57 and the region upstream of that gene were required for replication and stable maintenance and did not result in incompatibility.

Thank you, we have inserted this citation in the Methods section on line 337.

Referee #3:

In this manuscript, the authors investigated whether cp32 plasmids are required for the viability of Borrelia in both mice and ticks, and demonstrated that cp32 plasmids were dispensable for transmission of Borrelia between mice and ticks. This study is important as it gives additional evidence for redundancies among adhesive surface lipoproteins. In addition, generating the

Δ cp32 strain involved tremendous effort and provides an invaluable tool for investigating the role of prophages and the function of cp32-encoded lipoproteins. However, critical experiments presented in this manuscript appear to be incomplete as described below. If obtaining more complete data is not feasible, I strongly recommend that the authors acknowledge and discuss each of these issues in the text.

We thank Referee #3 for their careful review of our manuscript. We appreciate the Referee noting the importance of this study in demonstrating the infectivity of cp32-lacking *B. burgdorferi* and highlighting the future use of this derivative to determine the *in vivo* roles of cp32-encoded genes. Therefore, we have highlighted the major finding that cp32-lacking *B. burgdorferi* are able to transit the experimental infectious cycle and have further minimized the shortcomings of this manuscript to emphasize the major finding. We appreciate the helpful, constructive comments and have addressed them in full below.

Major points:

1 It is currently understood that *B. burgdorferi* outer surface proteins have multifaceted and redundant roles in complement component interactions as well as tissue targeted adhesion and colonization. This information seems critical to understanding the authors results and so should be featured prominently in the introduction.

We have added this to the Introduction on lines 107-109.

2 The introduction should also carefully summarize the past results on cp32-encoded lipoproteins including how those studies were done, rather than making passing mention of those studies in the results. This could help identify potential sources of discrepancies and provide clearer context for the current observations. For instance, the transposon mutant strain background used in Li et al. also lacks lp28-4, which encodes a few factors implicated in virulence including lipoproteins and complement regulating factors (e.g. WP_010890330.1). This raise the possibility that the transposon mutants decreased infectivity could be a result of the absence of lipoproteins encoded on lp28-4 as well as one or more cp32.

We have added a description of previous work on cp32-encoded lipoproteins to the Introduction on Lines 90-104.

3 The authors are encouraged to design and perform their experiments so that the outcomes are comparable to the past studies.

We agree that more *in vivo* work needs to be performed on these strains. Unfortunately, this work will not be able to be completed in time for this manuscript, yet we plan to perform more *in vivo* studies in the future. However, our main goal was to report the ability of cp32-lacking *B. burgdorferi* to survive the experimental mouse-tick infectious cycle. Therefore, we have revised the manuscript to reflect this.

3.A Li et al. identified transposon mutants of bbm38 and bbo39 (encoding ErpK and ErpM, respectively) be potentially important for infectivity. They examined five tissue types (ear, heart, joints, inoculation site (skin), and bladder) at two time points (2- and 4- week post-infection). The authors only examined two tissue types (ear and heart) at only one late time point (5-weeks post-infection).

Thank you for referencing this study, while similar studies on the Δ cp32 derivative won't be included in this manuscript, we plan to perform additional mouse-tick experiments and analyze additional tissues at earlier time points.

3.B In particular, the authors are urged to examine infectious phenotypes early postinfection since cp32 encoding gene products are suggested to be involved in innate immune evasion. Analyzing only 5-week specimens (after establishment of infection) could minimize or obscure any effect that takes place early.

Thank you for your suggestion, we really appreciate the input. Unfortunately, we are not able to include new studies in this manuscript, but are planning to design planned future experiments to better replicate previous experiments for comparable results.

4 The authors appear to focus on an overly restrictive hypothesis on how redundancies operate in cp32 encoded lipoproteins and also only examined transcript levels of a set of lipoproteins whose binding targets and functions are reported to be overlapping (Figure 3). It is possible that the regulation does not occur at the transcript level - for instance, they might occur through post-translational modifications. In addition, when the bacteria are challenged by the host immune response, redundant functions may not be due to the most similar proteins, but could be from factors functioning in parallel pathways facilitating innate immune evasion. Specifically, there are a number of adhesive surface lipoproteins reported in *Borrelia* including OspC, RevB, and VlsE. Investigating the hypothesized redundancy may require using a tool that has not currently been established (e.g. multiple target CRISPRi in vivo), but at least the authors should expand their discussion of how lipoprotein redundancy may influence innate immune evasion in *Borrelia*.

We have revised this paragraph on Lines 225-236 of the Discussion to better address this concern.

5 It is highly recommended for the number of the experimental animals used for wildtype and Δ cp32 to be more similar. In Figure 2B, there appears to be one data point for heart in Δ cp32. Similarly, there appears to be one data point for ear in wild-type in Figure 2C. Variable samples numbers, especially when so small, lowers the validity of the statistical comparisons.

Unfortunately, DNA was not extracted from every infected mouse and not all DNA extractions were successful. Therefore, in some instances, we only had usable DNA for qPCR from one or two mice. While we plan to repeat these experiments with more rigorous analysis, we currently do not have more infected mouse tissues for this manuscript. Our main objective for this manuscript is to reflect the fact that cp32-lacking *B. burgdorferi* are able to survive the experimental mouse-tick infectious cycle. We have revised the manuscript to better reflect this objective.

6 In Figure 3, bb0347 in nymph and *Imp1* in hearts were not analyzed. *Imp1* is also missing from Table 1.

We could not detect bb0347 transcript in either wt or Δ cp32-infected nymphs, which is described on lines 187-189 of the **Results**. Thank you for mentioning *Imp1*, which we intended to omit as we could not detect it in murine tissues and has now been removed from Figure 3.

Minor points:

1 Figure 2D, life stage (2) Unfed nymph, there is a two orders of magnitude difference in mean between wild-type and Δ cp32 strains and there generally seems to be more spirochete burden in Δ cp32 than in wild-type unfed nymph. In the text, the authors stated "The spirochete burdens in replete larvae and unfed nymphs were also similar between *I. scapularis* larvae infected with wt and Δ cp32 *B. burgdorferi*" - this seems like it could be statistical insignificance due to low samples numbers and thus is at best weak evidence.

Thank you for bringing this to our attention. When we analyzed all time points separately, wt-infected and Δ cp32-infected unfed nymphs were significantly different by the Mann-Whitney test. We have added this to the manuscript and fixed the Results on Lines 162-165 to reflect this.

2 "The ability of *B. burgdorferi* lacking all cp32 plasmids to complete the experimental mouse-tick infectious cycle raises the question as to why these plasmids are conserved in all sequenced Lyme disease *Borrelia* (Mongodin et al., 2013)? Perhaps the cp32 plasmids are not essential to complete the enzootic cycle in the lab environment but confer an unidentified selective advantage in nature." - This argument can be strengthened by noting the association of cp32 variant plasmids with disease phenotypes. The recent genome sequencing of clinical non-clinical isolates of Lyme-causing *Borrelia* found a significant association of cp32-4 and cp32-11 with dissemination (Lemieux 2023 (doi: 10.1101/2023.02.26.530159)).

Thank you for calling this to our attention. We have added a sentence describing this on Lines 256-258 of the Discussion.

3 There is a typo on page 4, line 21. Serconversion -> Seroconversion.

Thank you. We have fixed this typo, now on line 154.

4 The following statement from the discussion: "The in vivo data from this study, as well as published data cited in Table 1, demonstrate the shortcomings of using in vitro binding assays to infer the in vivo functions of genes." seems needlessly combative as it only shows the shortcomings of overinterpretation of in vitro data. Especially given the known redundancy of adhesive lipoproteins, no observed negative effect from absence of cp32 and no statistically significant differences in transcripts from a subset of the non-cp32 genes does not demonstrate that the interaction detected in in vitro studies does not occur in vivo.

We have re-worded lines 237-251 of this paragraph to be less combative.

Dear Dr. Wachter,

Thank you for the submission of your revised manuscript to our editorial offices. I have already forwarded to you the reports from three referees that I asked to re-evaluate your study, you will find again below.

As you know, referee #2 is satisfied by the revision. In contrast, referees #1 and #3 still consider the work as preliminary and indicate that additional experiments that support the conclusions and verify the reproducibility of the observations are required. As both critical referees indicated that they support a further revision of the study and after discussing with you, I ask you to address the remaining concerns in a final revised manuscript. Please also provide a detailed final p-b-p-response to the referee points.

Moreover, I have these editorial requests I ask you to address:

- Please provide a title with not more than 100 characters (including spaces).
- Please provide the abstract written in present tense throughout.
- We updated our journal's competing interests policy in January 2022 and request authors to consider both actual and perceived competing interests. Please review the policy <https://www.embopress.org/competing-interests> and add a section regarding your competing interests to the manuscript. Please name this section 'Disclosure and Competing Interests Statement' and put it after the Acknowledgements section.
- Please make sure that all the funding information is also entered into the online submission system and that it is complete and similar to the one in the acknowledgement section of the manuscript text file.
- Regarding data quantification and statistics, please make sure that the number "n" for how many independent experiments were performed, their nature (biological versus technical replicates), the bars and error bars (e.g. SEM, SD) and the test used to calculate p-values is indicated in the respective figure legends (main and EV figures). Please also check that all the p-values are explained in the legend, and that these fit to those shown in the figure. Please provide statistical testing where applicable. Please avoid the phrase 'independent experiment', but clearly state if these were biological or technical replicates. Please also indicate (e.g. with n.s.) if testing was performed, but the differences are not significant. In case n=2 please show the data as separate datapoints or bars without error bars and statistics. See also: <http://www.embopress.org/page/journal/14693178/authorguide#statisticalanalysis>

- Please make sure that all figure panels are called out separately and sequentially. Presently, there are no separate panel callouts for Fig. 3A-Q. Please check.
- I would suggest to provide the information contained in Supplementary Table S1 as a reagents and tools table. I have attached templates for that in word or excel format. Please upload the filled in table to the manuscript tracking system as a new 'Reagent Table' file. Please also adjust any callouts to this table. The example linked below shows how the table will display in the published article and includes examples of the type of information that should be provided for the different categories of reagents and tools. Please list your reagents/tools using the categories provided in the template and do not add additional subheadings to the table. Reagents/tools that do not fit in any of the specific categories can be listed under "Other": https://www.embopress.org/pb%2Dassets/embo-site/msb_177951_sample_FINAL.pdf
- Please also provide the source data for any new data/figure added to the final revised manuscript.
- Finally, please find also attached a word file of the manuscript text (provided by our publisher) with changes we ask you to include in your final manuscript text and comments. Please use the attached file as basis for further revisions and provide your final manuscript file with track changes, in order that we can see any modifications done.

In addition, I would need from you:

- a short, two-sentence summary of the manuscript (not more than 35 words).
- two to four short bullet points highlighting the key findings of your study (two lines each).
- a schematic summary figure as separate file that provides a sketch of the major findings (not a data image) in jpeg or tiff format (with the exact width of 550 pixels and a height of not more than 400 pixels) that can be used as a visual synopsis on our website.

Yours sincerely,

Referee #1:

I appreciate the authors' efforts to revise the manuscript. Unfortunately, the major issues remain, which are insufficient number of animals and tissues per animal (this has been mentioned by me in the point 2 and 3, and the reviewer 3 (see point 5 of the previous comments from the reviewer 3).

Basically, the bacterial infectivity is defined by the behavior/outcomes of bacteria after infection. Compared to the wild-type bacterial strain, the scenarios of a bacterial mutant strain displaying inability to colonize tissues, lower ability to colonize tissues, or preference to colonize specific tissues are all considered "altered infectivity." In this manuscript, the statistically unaltered bacterial burdens in tested tissues are based on very small number of animals (n=2 in some cases). However, if the number of animals included in the manuscript increase, there may be a possibility that the statistical differences of bacterial burdens are detected in those tissues, resulting in the conclusion of "altered infectivity." Additionally, as mentioned in the previous comments, some genes encoded on the cp32 plasmids have been documented to confer tropism to a particular tissue, altering the infectivity. Therefore, the results of unaltered colonization in this manuscript could simply because insufficient tissues were collected. In other words, if more tissues are included, it is very likely that the differences of bacterial burdens in those additional tissues would differ, changing the conclusion of "unaltered infectivity." Therefore, in the above-mentioned cases, the current finding of "unaltered infectivity" may be very likely due to the insufficient number of animals and tissues per animal. Thus, adding more animals or tissues per animal would be required in order to make the conclusion reflect the title.

As I mentioned in my previous comments, this manuscript is impactful and exciting because the authors tested whether the infectivity is altered when Lyme borreliae are missing all cp32 plasmids. I understand that the authors may try to downplay the impact of the paper by narrowing the scope of the manuscript to test whether cp32 plasmids are required for survival in vivo. However, such downplaying would significantly decrease the impacts mentioned above. The information that could be provided would thus be very limited. I would strongly suggest the authors to add more animals or tissues per animal.

Referee #2:

Upon review of the revised manuscript submitted by Wachter et. al., I believe that the authors have adequately addressed the major concerns and criticisms of the initial draft. In particular, they have sufficiently addressed my specific comments to the extent to which they were able. While I had requested further data relating to the growth of individual plasmid-knock outs, I also understand the technical and temporal limitations with growing and analyzing a fastidious organism. I think that the experiments and analysis proposed by me and the other reviewers would strengthen the overall conclusions of the paper and may add nuance to our understanding of the role that cp32s may play in *Borrelia*, however I also feel that the primary finding that a cp32-less *B. burgdorferi* can complete the enzootic cycle is a significant enough finding to stand alone and warrants publication. Presumably the other analyses will be the backbone of additional follow-up on this interesting and important observation.

Referee #3:

Wachter et al. studied the infectivity of *Borrelia* lacking all cp32 in mice and ticks. This study has good potential to yield significant insights, but is currently incomplete since it lacks strong evidence for the conclusions drawn about murine infection. Therefore, additional experiments that support the conclusions and verify the reproducibility of the observations are required. Since none of the follow-up experiments could be performed within the revision period, I think it would be worthwhile to resubmit this work after completing the experiments and doing a more thorough analysis. If the authors decide to try to publish this manuscript with the current dataset, I strongly advise them to revise the text much more rigorously by replacing all statements that are not supported by the current data (details below) with more justifiable interpretations.

Major points

1. Insufficient data.

The revised manuscript contains neither additional data nor analyses to fill in critical gaps raised by reviewers. Without these changes, the manuscript lacks the completeness and thoroughness expected for publication. In particular, the assessment of

spirochete loads in mice was only done once and with insufficient sample numbers. In addition, the removal of some of the unsupported conclusions from the original manuscript has left very few new insights; the re-presentation of a significant part of Figure 1 from the previous manuscript further dilutes the novelty of this study. These aspects render this study inconclusive and unconvincing in its current form.

1A. Following the sentences first mentioning Figure 2B and C, which say that the spirochete loads were analyzed (quoted below), there is no text describing the analysis.

(156-157) "Additionally, the spirochete loads in tissues (ear and heart) of needle-inoculated mice were analyzed (Fig 2B)."
(174-175) "Spirochete loads in tissues (ear and heart) were also analyzed for mice infected with wt or Δ cp32 spirochetes by tick bite (Fig 2C)."

The authors are advised to provide clear descriptions and interpretations of their results from the figures when they are first referred to in the text.

1B. Significant parts of Figure 1 and the corresponding text were already presented in Wachter et al. 2023. It is appropriate to give brief background information from the previous study, however, when the main conclusions are general and weak, re-iterating the pre-published methods by both text and figures dilutes the novelty of the study. The following text from Wachter 2023 is more-or-less duplicated in this manuscript (lines 130-144 - refer to Figure 1A-B).
"... we displaced the entire set of cp32 plasmids from both wt and flacp::ibbd18 spirochetes. This was accomplished through sequential rounds of transformation with 7 distinct shuttle vectors (SV). Each SV contained a fragment of B31 cp32 DNA encompassing a partial set of plasmid replication and partition genes, extending from nucP to the PFam 32 gene, from a single cp32 plasmid (e.g. bbp29-bbp32 of cp32-1) (Fig. 7a). These replication-competent SVs were incompatible with their cognate plasmid, thus displacing the cp32 from which they were derived. However, they were unstable due to omission of the parB/Pfam 49 gene and readily lost when antibiotic selection was lifted. After a clonal transformant was confirmed to lack both the targeted cp32 and the displacing SV, it was subjected to another round of transformation with a different SV targeting a remaining cp32. This process was repeated until all 7 cp32 plasmids had been sequentially displaced from both wt and flacp::ibbd18 cells (Fig. 7b)".

2. The main conclusion is not well supported.

All revisions appear to be done at the level of wording changes and elimination of some inappropriate data from figures. Since too few tissue samples were analyzed to give the comparison between Δ cp32 and wild-type statistical significance, the only conclusion that can be made is that *Borrelia* lacking all cp32 plasmids can infect *Mus musculus*. Despite the authors' acknowledgement of this point, the revised manuscript still contains unjustified statements comparing mouse infectivity between Δ cp32 and wild-type strains.

The following statements are not supported by the current dataset.

2A. (120-122) "Herein, we report that *B. burgdorferi* lacking all cp32 plasmids can successfully complete the infectious cycle, **with no defect** in transmission and infection of the murine host."

"No defect" is an overstatement since the analyses are incomplete. The small number of tissue samples examined in murine infection and failure to examine early infection stages undermines this statement.

2B. (224-225) " **Δ cp32 spirochetes were as infectious as wt** cells when compared ... by spirochete isolation at 5 weeks..."
As above, there is not enough data for a meaningful quantitative comparison.

2C. (234-236) "... our findings conclusively demonstrate that **cp32-encoded gene products are not required at any point of the experimental laboratory mouse-tick infectious cycle.**"

Since the authors did not examine early post-infection when innate immune evasion has a primary role, the preceding statement is misleading.

2D. (lines 1-2) "**Unaltered** infectivity of the Lyme disease spirochete, *Borrelia burgdorferi*, lacking all cp32 prophage plasmids ..."

The only conclusion supported by the data is that *Borrelia* lacking all cp32 can infect mice and ticks. Since the degree of Δ cp32 infectivity relative to wild-type in mice remains unknown due to an inadequate sample size, the use of the word "Unaltered" is misleading.

2E. (781-783) "**Similar numbers of wt and Δ cp32 genomes were detected by qPCR in ear and heart tissues of mice** infected with either strain by needle inoculation (B) or tick bite (C)."

As above, there is not sufficient data for a meaningful quantitative comparison.

Minor point:

There is a section titled "Results and Discussion" (line 129), which now contains a "Discussion" sub-heading (line 191). In general, when the results and discussion sections are combined in a short article, each result is discussed in turn rather than keeping the two sections separate.

Referee #1:

I appreciate the authors' efforts to revise the manuscript. Unfortunately, the major issues remain, which are insufficient number of animals and tissues per animal (this has been mentioned by me in the point 2 and 3, and the reviewer 3 (see point 5 of the previous comments from the reviewer 3).

Basically, the bacterial infectivity is defined by the behavior/outcomes of bacteria after infection. Compared to the wild-type bacterial strain, the scenarios of a bacterial mutant strain displaying inability to colonize tissues, lower ability to colonize tissues, or preference to colonize specific tissues are all considered "altered infectivity." In this manuscript, the statistically unaltered bacterial burdens in tested tissues are based on very small number of animals (n=2 in some cases). However, if the number of animals included in the manuscript increase, there may be a possibility that the statistical differences of bacterial burdens are detected in those tissues, resulting in the conclusion of "altered infectivity. Additionally, as mentioned in the previous comments, some genes encoded on the cp32 plasmids have been documented to confer tropism to a particular tissue, altering the infectivity. Therefore, the results of unaltered colonization in this manuscript could simply because insufficient tissues were collected. In other words, if more tissues are included, it is very likely that the differences of bacterial burdens in those additional tissues would differ, changing the conclusion of "unaltered infectivity." Therefore, in the above-mentioned cases, the current finding of "unaltered infectivity" may be very likely due to the insufficient number of animals and tissues per animal. Thus, adding more animals or tissues per animal would be required in order to make the conclusion reflect the title.

As I mentioned in my previous comments, this manuscript is impactful and exciting because the authors tested whether the infectivity is altered when Lyme borreliæ are missing all cp32 plasmids. I understand that the authors may try to downplay the impact of the paper by narrowing the scope of the manuscript to test whether cp32 plasmids are required for survival in vivo. However, such downplaying would significantly decrease the impacts mentioned above. The information that could be provided would thus be very limited. I would strongly suggest the authors to add more animals or tissues per animal.

We thank Referee #1 for their continued thoughtful feedback. We sincerely appreciate the time you have dedicated to reviewing our manuscript and your constructive suggestions.

We have conducted additional animal experiments to significantly increase the sample size for needle inoculated mice in response to your concerns. We apologize for the length of time it took to complete these experiments. However, the repeated studies still showed no significant differences in bacterial infectivity and tissue load between the wild-type and cp32-less strains.

We understand the importance of sufficient sample sizes in detecting subtle differences in infectivity and appreciate your point regarding the potential for altered infectivity to be masked by small sample sizes. Therefore, in addition to adding more mice, we also added two more time points 2- and 4-weeks post needle inoculation and added joints in our tissue load analysis.

We thank you for your insights on the scope of this manuscript and agree that this study addresses an exciting and impactful question about the role of the cp32 plasmids in *Lyme borreliae* infectivity.

We greatly appreciate your input and have revised the manuscript to address your concerns. We would like to, once again, thank you for your constructive feedback.

Referee #2:

Upon review of the revised manuscript submitted by Wachter et. al., I believe that the authors have adequately addressed the major concerns and criticisms of the initial draft. In particular, they have sufficiently addressed my specific comments to the extent to which they were able. While I had requested further data relating to the growth of individual plasmid-knock outs, I also understand the technical and temporal limitations with growing and analyzing a fastidious organism. I think that the experiments and analysis proposed by me and the other reviewers would strengthen the overall conclusions of the paper and may add nuance to our understanding of the role that cp32s may play in *Borrelia*, however I also feel that the primary finding that a cp32-less *B. burgdorferi* can complete the enzootic cycle is a significant enough finding to stand alone and warrants publication. Presumably the other analyses will be the backbone of additional follow-up on this interesting and important observation.

We thank Referee #2 for their careful review of the manuscript and the encouraging feedback. We greatly appreciate the recognition of our efforts. While we have increased the number of mice for the revised draft, we are committed to continuing research on these cp32-less *B. burgdorferi*. We would like to once again, thank you for the valuable feedback throughout this review process.

Referee #3:

Wachter et al. studied the infectivity of *Borrelia* lacking all cp32 in mice and ticks. This study has good potential to yield significant insights, but is currently incomplete since it lacks strong evidence for the conclusions drawn about murine infection. Therefore, additional experiments that support the conclusions and verify the reproducibility of the observations are required. Since none of the follow-up experiments could be performed within the revision period, I think it would be worthwhile to resubmit this work after completing the experiments and doing a more thorough analysis. If the authors decide to try to publish this manuscript with the current dataset, I strongly advise them to revise the text much more rigorously by replacing all statements that are not supported by the current data (details below) with more justifiable interpretations.

We thank Referee #3 for their continued careful review of our manuscript. We apologize for the time it took to add additional mouse studies but believe that we have made this work stronger and have validated our initial findings. We have addressed the issues raised below.

Major points

1. Insufficient data.

The revised manuscript contains neither additional data nor analyses to fill in critical gaps raised by reviewers. Without these changes, the manuscript lacks the completeness and thoroughness expected for publication. In particular, the assessment of spirochete loads in mice was only done once and with insufficient sample numbers. In addition, the removal of some of the unsupported conclusions from the original manuscript has left very few new insights; the re-presentation of a significant part of Figure 1 from the previous manuscript further dilutes the novelty of this study. These aspects render this study inconclusive and unconvincing in its current form.

We have repeated the needle-inoculated murine experiments with significantly more mice, analyzed more tissues, and included more time points. By adding statistically significant numbers of mice, we can conclude that the loss of the cp32 plasmids does not affect infectivity or tissue load in ears, hearts, and joints at 2-, 4-, and 5-weeks post needle inoculation.

1A. Following the sentences first mentioning Figure 2B and C, which say that the spirochete loads were analyzed (quoted below), there is no text describing the analysis.

(156-157) "Additionally, the spirochete loads in tissues (ear and heart) of needle-inoculated mice were analyzed (Fig 2B)."

Thank you for pointing this out. This is now contained in lines 169-173 and we have fixed this to indicate that qPCR was utilized to determine tissue loads.

"Additionally, the spirochete loads in tissues (ears, hearts, and joints) of needle-inoculated mice were analyzed by qPCR at 2-, 4-, and 5- weeks post-inoculation, with no significant differences in tissue load determined between wt and Δ cp32 spirochetes at any time point (Fig 2B and Supplemental Fig S2)."

(174-175) "Spirochete loads in tissues (ear and heart) were also analyzed for mice infected with wt or Δ cp32 spirochetes by tick bite (Fig 2C)."

Thank you, this sentence is now on lines 190-191. We have also fixed this sentence to indicate that it was analyzed by qPCR.

"Spirochete loads in tissues (ear and heart) were also analyzed by qPCR for mice infected with wt or Δ cp32 spirochetes by tick bite at 5-weeks post feeding (Fig 2C)."

The authors are advised to provide clear descriptions and interpretations of their results from the figures when they are first referred to in the text.

1B. Significant parts of Figure 1 and the corresponding text were already presented in Wachter et al. 2023.

It is appropriate to give brief background information from the previous study, however, when the main conclusions are general and weak, re-iterating the pre-published methods by both text and figures dilutes the novelty of the study. The following text from Wachter 2023 is more-or-less duplicated in this manuscript (lines 130-144 - refer to Figure 1A-B).

"... we displaced the entire set of cp32 plasmids from both wt and flacp::ibbd18 spirochetes. This was accomplished through sequential rounds of transformation with 7 distinct shuttle vectors

(SV). Each SV contained a fragment of B31 cp32 DNA encompassing a partial set of plasmid replication and partition genes, extending from nucP to the PFam 32 gene, from a single cp32 plasmid (e.g. bbp29-bbp32 of cp32-1) (Fig. 7a). These replication-competent SVs were incompatible with their cognate plasmid, thus displacing the cp32 from which they were derived. However, they were unstable due to omission of the parB/Pfam 49 gene and readily lost when antibiotic selection was lifted. After a clonal transformant was confirmed to lack both the targeted cp32 and the displacing SV, it was subjected to another round of transformation with a different SV targeting a remaining cp32. This process was repeated until all 7 cp32 plasmids had been sequentially displaced from both wt and flacp::ibbd18 cells (Fig. 7b)".

Thank you for addressing this, we did not copy any figures from our previous publication in Figure 1. All the images presented in this manuscript were created for this publication. While we do reference back to our initial paper and state that the work was done and published in our previous manuscript, we believe that it is important to briefly re-state the work that was performed to generate the *B. burgdorferi* derivative being studied in this manuscript.

2. The main conclusion is not well supported.

All revisions appear to be done at the level of wording changes and elimination of some inappropriate data from figures. Since too few tissue samples were analyzed to give the comparison between Δ cp32 and wild-type statistical significance, the only conclusion that can be made is that *Borrelia* lacking all cp32 plasmids can infect *Mus musculus*. Despite the authors' acknowledgement of this point, the revised manuscript still contains unjustified statements comparing mouse infectivity between Δ cp32 and wild-type strains.

We have addressed this issue by adding additional numbers of needle-inoculated mice to provide statistical significance to our finding.

The following statements are not supported by the current dataset.

2A. (120-122) "Herein, we report that *B. burgdorferi* lacking all cp32 plasmids can successfully complete the infectious cycle, **with no defect** in transmission and infection of the murine host." "No defect" is an overstatement since the analyses are incomplete. The small number of tissue samples examined in murine infection and failure to examine early infection stages undermines this statement.

We have addressed this by adding significant numbers of needle-inoculated mice, adding additional time points following needle inoculation, and analyzing significant numbers of ears, hearts, and joints at each time point. This is now contained in lines 121-123. We did not change this statement as we have added additional mice, tissues, and time points analyzed and still did not detect any differences between wt and cp32-less *B. burgdorferi*.

2B. (224-225) " **Δ cp32 spirochetes were as infectious as wt** cells when compared ... by spirochete isolation at 5 weeks..."

As above, there is not enough data for a meaningful quantitative comparison.

In concatenating the “Results” and “Discussion” sections, we have removed this from our manuscript. However, this statement is still supported with the addition of more mice, tissues, and time points.

2C. (234-236) "... our findings conclusively demonstrate that **cp32-encoded gene products are not required at any point of the experimental laboratory mouse-tick infectious cycle.**" Since the authors did not examine early post-infection when innate immune evasion has a primary role, the preceding statement is misleading.

Thank you for pointing this out. We have added an earlier 2-week time point and do not see any difference in tissue load in ears, hearts, and joints even at this earlier time point. As this remains true with additional data, this sentence was retained and is now on lines 227-228.

2D. (lines 1-2) "**Unaltered** infectivity of the Lyme disease spirochete, *Borrelia burgdorferi*, lacking all cp32 prophage plasmids ..."

The only conclusion supported by the data is that *Borrelia* lacking all cp32 can infect mice and ticks. Since the degree of Δ cp32 infectivity relative to wild-type in mice remains unknown due to an inadequate sample size, the use of the word "Unaltered" is misleading.

As we have added statistically significant numbers of needle-inoculated mice, including analysis of their tissues, at additional time points, we have kept the title.

2E. (781-783) "**Similar numbers of wt and Δ cp32 genomes were detected by qPCR in ear and heart tissues of mice** infected with either strain by needle inoculation (B) or tick bite (C)." As above, there is not sufficient data for a meaningful quantitative comparison.

This is now contained in lines 698-700. As we now have significant data for needle-inoculated mice, we have kept this statement in Figure 2.

Minor point:

There is a section titled "Results and Discussion" (line 129), which now contains a "Discussion" sub-heading (line 191). In general, when the results and discussion sections are combined in a short article, each result is discussed in turn rather than keeping the two sections separate.

Thank you for the advice! We have fixed this issue and have put the discussion after the results. However, we have kept our hypotheses and conclusions at the end (229-265).

Dear Dr. Wachter,

Thank you for the submission of your further revised manuscript to our editorial offices. I have now received the reports from two of the three referees that I asked to re-evaluate the study, you will find below. As you will see, both referees now fully support the publication of the study in EMBO reports. However, both referees have remaining concerns and suggestions to improve the manuscript, I ask you to address in a final revised manuscript. Please also provide a final p-b-p-response to the referee points.

- Please provide a final, more comprehensive title (with not more than 100 characters - including spaces) highlighting the main findings in a more active way.
- We updated our journal's competing interests policy in January 2022 and request authors to consider both actual and perceived competing interests. Please review the policy <https://www.embopress.org/competing-interests> and add a section regarding your competing interests to the manuscript. Please name this section 'Disclosure and Competing Interests Statement' and put it after the Acknowledgements section.
- We now use CRediT to specify the contributions of each author in the journal submission system. CRediT replaces the author contribution section. Please use the free text box to provide more detailed descriptions. Thus, please do NOT provide your final manuscript text file with an author contributions section. See also guide to authors: <https://www.embopress.org/page/journal/14693178/authorguide#authorshipguidelines>
- We request that primary datasets produced in a study (e.g. RNA-seq, ChIP-seq, structural and array data) are deposited in an appropriate public database. If no primary datasets have been deposited, please also state this in a dedicated data availability section (e.g. 'No primary datasets have been generated and deposited').
- The Expanded View format, which will be displayed in the main HTML of the paper in a collapsible format, has replaced the Supplementary information. You can submit up to 5 images as Expanded View. Please follow the nomenclature Figure EV1, Figure EV2 etc. The figure legend for these should be included in the main manuscript document file in a section called Expanded View Figure Legends after the main Figure Legends section.
- Please order the manuscript sections like this, using these names:
Title page - Abstract - Keywords - Introduction - Results & Discussion - Methods - Data availability section - Acknowledgements (including funding information) - Disclosure and Competing Interests Statement - References - Figure legends - Expanded View Figure legends
- Please make sure that all the funding information is also entered into the online submission system and that it is complete and similar to the one in the acknowledgement section of the manuscript text file. The information is still not congruent: VIDO needs to be entered as a separate funder (or Ministry of Agriculture and from the Canada Foundation for Innovation) and the sentence from the Comments box needs to be removed since VIDO needs to be entered separately.
- Please make sure that all figure panels are called out separately and sequentially. Presently, there are no separate panel callouts for Fig. 3A-Q. Please check.
- Please change the callout of Supplementary Table S1 to 'reagents and tools table'. Moreover, please use our new templates (.doc) for the Reagents and Tools Table that can be found in our author guidelines (section 'Structured Methods'):
<https://www.embopress.org/page/journal/14693178/authorguide#manuscriptpreparation>
- Please acknowledge BioRender with a paragraph (named Graphics) in the Methods section, indicating which panels/objects were created using BioRender.com. Then, please remove all the other mentions of BioRender from the manuscript text file.
- Thank you for providing the requested source data. Please upload this as one folder per figure (with all files for one figure in one folder and ZIPed together) and one folder for all the source data for the EV figures.

Please let me know if you have questions regarding the revision.

Yours sincerely,

Achim Breiling

Referee #1:

I appreciate the authors' efforts to add more animals for the work and am satisfied for the quality of the work.

The only minor concern is that qPCR results of Supplemental Fig. 2 for the bacterial burdens are supposed to be plotted as a graph with a table beneath that. However, a current supplemental Fig. 2 seems to be an agarose gel, which seems to be not related to this work. The authors may need to correct that and place the correct figure on the supplemental Fig. 2.

Referee #3:

Hillman et al. studied the infectivity of *Borrelia* lacking all cp32 in mice and ticks. In this revised manuscript, the authors performed the recommended animal infection experiments with an increased number of samples. Their result convincingly showed that the absence of cp32s does not affect the infectivity of *Borrelia* at 5-weeks post infection. Their work is worth publication; however, the current manuscript contains mistakes and issues that need to be corrected.

Major points

1. The authors stated that "the spirochete loads in tissues (ears, hearts, and joints) of needle-inoculated mice were analyzed by qPCR at 2-, 4-, and 5- weeks post-inoculation, with no significant differences in tissue load..."(lines 171-174). In this manuscript, the main figure only contains the data for 5-week post-infection (Fig 2A and B), but not for the earlier timepoints. Since many surface lipoproteins encoded in cp32s are reported to be involved in host innate immune evasion and thus to be important in early infection, I think that the spirochete load 2-weeks post infection in mice should also be in the main figure (with the 5-week post infection data) rather than in the supplemental figure.
2. The current Fig S2 does not match the descriptions in the main text and figure legends. The figure legend indicates that Fig S2 shows the "infectivity of wt and Δ cp32 spirochetes in mice at 2- and 4-weeks" (line 759-760). However, the current supplemental Fig S2 shows a raw gel image of PCR products similar to Fig S1, which does not correspond to the main text (lines 171-174).

Minor points

1. The heading "In vivo gene expression" (line 208) is too broad to be meaningful and needs to be more descriptive.
2. The following sentence is incomprehensible: "To determine infection and spirochete load, tissues were harvested at 2-, 4-, and at 5- weeks (following larval feeding) post-injection through and attempted isolation of spirochetes from mouse tissues (ear, bladder, joint, and fat) was attempted in liquid culture along with DNA and RNA extraction of mouse tissues (ear, heart, and joint)" (lines 351-354).

Referee #1:

I appreciate the authors' efforts to add more animals for the work and am satisfied for the quality of the work.

The only minor concern is that qPCR results of Supplemental Fig. 2 for the bacterial burdens are supposed to be plotted as a graph with a table beneath that. However, a current supplemental Fig. 2 seems to be an agarose gel, which seems to be not related to this work. The authors may need to correct that and place the correct figure on the supplemental Fig. 2.

We would like to thank Referee #1 for their careful review of our manuscript, especially given the length of time it took between resubmissions. We also greatly appreciate the correction of our mistake. The incorrect Supplemental Figure 2 has been replaced with the correct Figure (now called Fig EV2), which contains the spirochete load in murine tissues at 4-weeks post inoculation. We have taken the data at 2-weeks post inoculation from Supplemental Figure 2 and placed it in Figure 2 as suggested by Referee #3.

Referee #3:

Hillman et al. studied the infectivity of *Borrelia* lacking all cp32 in mice and ticks. In this revised manuscript, the authors performed the recommended animal infection experiments with an increased number of samples. Their result convincingly showed that the absence of cp32s does not affect the infectivity of *Borrelia* at 5-weeks post infection. Their work is worth publication; however, the current manuscript contains mistakes and issues that need to be corrected.

We would like to thank Referee #2 for their careful review of our manuscript, especially considering the time elapsed between resubmissions. We have addressed the major and minor points below.

Major points

1. The authors stated that "the spirochete loads in tissues (ears, hearts, and joints) of needle-inoculated mice were analyzed by qPCR at 2-, 4-, and 5- weeks post-inoculation, with no significant differences in tissue load..."(lines 171-174). In this manuscript, the main figure only contains the data for 5-week post-infection (Fig 2A and B), but not for the earlier timepoints. Since many surface lipoproteins encoded in cp32s are reported to be involved in host innate immune evasion and thus to be important in early infection, I think that the spirochete load 2-weeks post infection in mice should

also be in the main figure (with the 5-week post infection data) rather than in the supplemental figure.

We recognize the importance of similar tissue loads at early time-points. Accordingly, we have amended Figure 2 to contain data for both 2-week and 5-week post infection time points in mice. This statement is now on lines 174-177.

2. The current Fig S2 does not match the descriptions in the main text and figure legends. The figure legend indicates that Fig S2 shows the "infectivity of wt and Δ cp32 spirochetes in mice at 2- and 4-weeks" (line 759-760). However, the current supplemental Fig S2 shows a raw gel image of PCR products similar to Fig S1, which does not correspond to the main text (lines 171-174).

We greatly appreciate being notified of this mistake. The incorrect Supplemental Figure 2 has been replaced with the correct Figure (now called Fig EV2), which contains the spirochete load in murine tissues at 4-weeks post inoculation. We have taken the data at 2-weeks post inoculation from Supplemental Figure 2 and placed it in Figure 2 as suggested.

Minor points

1. The heading "In vivo gene expression" (line 208) is too broad to be meaningful and needs to be more descriptive.

We have changed the heading to now read "Gene expression of select genes *in vitro* and *in vivo*" (now on line 214).

2. The following sentence is incomprehensible: "To determine infection and spirochete load, tissues were harvested at 2-, 4-, and at 5- weeks (following larval feeding) post-injection through and attempted isolation of spirochetes from mouse tissues (ear, bladder, joint, and fat) was attempted in liquid culture along with DNA and RNA extraction of mouse tissues (ear, heart, and joint)" (lines 351-354).

Thank you for bringing this to our attention. We have changed this statement to read more clearly: "To determine infection and spirochete load, tissues were harvested at 2-, 4-, and 5-weeks (following larval feeding) post-injection. Spirochetes were isolated from mouse tissues (ear, bladder, joint, and fat) through liquid culture, and DNA and RNA were extracted from mouse tissues (ear, heart, and joint)" (now on lines 367-370).

Dr. Jenny Wachter
VIDO
120 Veterinary Rd
Saskatoon, SK S7N 5E3
Canada

Dear Dr. Wachter,

Thank you for the submission of your further revised manuscript to our editorial offices. I now went through your final p-b-p-response, I consider the remaining points of the referees as adequately addressed.

I am thus pleased to inform you that your manuscript has been accepted for publication in EMBO reports. Your manuscript will be processed for publication by EMBO Press. It will be copy edited and you will receive page proofs prior to publication. Please note that you will be contacted by Springer Nature Author Services to complete licensing and payment information.

Yours sincerely,
